# Factual and Tailored Recommendation Endorsements using Language Models and Reinforcement Learning

**Jihwan Jeong**[*]
University of Toronto

**Yinlam Chow**[†]
Google Research

**Guy Tennenholtz**
Google Research

**Chih-wei Hsu**
Google Research

**Aza Tulepbergenov**
Google Research

**Mohammad Ghavamzadeh**[*]
Amazon

**Craig Boutilier**
Google Research

## Abstract

Recommender systems (RSs) play a central role in matching candidate items to users based on their preferences. While traditional RSs rely on user feedback signals, conversational RSs interact with users in natural language. In this work, we develop P$^4$LM, an *aPpealing*, *Precise*, *Preference-comprehensive* and *Prioritized* language model which endorses recommended items by emphasizing specific item characteristics and their coverage to a user's preferences. P$^4$LM uses an *embedding* representation of a user's preferences to generate responses that are appealing, factually-grounded and tailored to the user's preferences. P$^4$LM employs a joint reward function to measure precision, appeal, preference coverage and prioritization of preferences, which are used as AI-based feedback in a reinforcement learning-based language model framework. On the MovieLens 25M and Amazon Product Review datasets, P$^4$LM delivers more appealing and tailored endorsements to users, as determined by auto-critic and rater evaluations.

## 1 Introduction

Recommender systems (RSs) have emerged as a dominant way in which users discover content, products, and services (Resnick & Varian, 1997). Traditional RSs match candidate items to users based on estimates of their item preferences. However, these estimated preferences are often based on user behavioral signals (e.g., clicks, consumption, ratings,

---

[*]Work performed while at Google Research.
[†]Corresponding author: yinlamchow@google.com.

purchases). Unfortunately, this provides little opportunity for an RS to elicit preference information from users or to explain its recommendations. *Conversational RSs* employ natural language to facilitate more effective communication between RSs and their users (Sun & Zhang, 2018; Lei et al., 2020; Shen et al., 2023).

The emergence of language models (LMs) as a powerful paradigm for user interaction (Li et al., 2018; Friedman et al., 2023) suggests their use in conversational RSs. However, this requires LMs to engage with users in a personalized manner, consistent with their preferences. In this paper, we explore the use of LMs to enrich the user's RS experience. We develop techniques that allow an LM to communicate the nuances of recommended items (as determined by a back-end RS) to a user, detailing their features and benefits, and *explaining their alignment with the user's preferences*. Such *personalized LMs* are not meant to "convince" users in the traditional sense, but rather to articulate the *genuine, relevant merits* of a recommended item relative to the user.

Although existing RS technologies are adept at predicting a user's favored items, a personalized LM can provide an enriched experience by tailoring suggestions to what a user genuinely needs and values. However, a number of challenges must be addressed in this endeavor: (i) the integrity and accuracy of an item's information is paramount; (ii) the LM's endorsement should be appealing; (iii) the LM should present a reasonably comprehensive portrayal of the item by articulating its merits and drawbacks, with a focus on *coverage* of the user's preferences; and finally (iv) the endorsement should prioritize its focus on user preferences based on aspects that matter most to the user. With these four criteria, RS endorsements can facilitate informed user decisions and thus improve user engagement. In this work, we develop an LM framework for RS endorsements centered on these principles.

Figure 1: $P^4$LM produces precise, appealing, comprehensive, and tailored endorsements from a user embedding and item description, as demonstrated with Toy Story. It relies solely on the user embedding, not requiring a textual user profile.

Our contributions are three-fold. First, we quantify the four criteria above to allow systematic evaluation. Second, leveraging recent advances in reinforcement learning (RL) from AI feedback (RLAIF) (Lee et al., 2023) and reward aggregation, we develop an LM fine-tuning methodology to better align an LM with the four criteria. The resulting $P^4$LM framework not only embodies semantic skill, but also understands user preferences as encoded by an RS embedding, providing factual, appealing, and tailored endorsements. Finally, using the MovieLens 25M (Harper & Konstan, 2015) and Amazon Product Review (Ni et al., 2019) datasets, we show that $P^4$LM can power RS endorsements that promote customized, relevant, and holistic interactions with users.

We begin with an introduction of RSs, and the endorsement task description in Sec. 2. We develop the supervised fine-tuned RS endorsement LM in Sec. 3. In Sec. 4, we describe $P^4$LM, our RL framework, which leverages AI feedback to optimize the four endorsement criteria. Finally, in Sec. 5 we demonstrate the effectiveness of $P^4$LM at generating factual, appealing, and user-tailored recommendation endorsements on the MovieLens 25M and Amazon Product Review datasets, using both automatic and rater evaluations.

## 2  Problem Formulation

We provide background on RSs and formulate the recommendation endorsement problem through our four principal criteria.

**Recommender Systems.**  To model user preferences over items in an RS, we assume a standard collaborative filtering (CF) model (Su & Khoshgoftaar, 2009), where user behavioral data (e.g., clicks, ratings, etc.) are used to learn user and item *representations* which are combined to predict a user's preference for a given item. We assume users $\mathcal{U}$, items $\mathcal{I}$, and a (usually sparse) $|\mathcal{I}| \times |\mathcal{U}|$ ratings matrix $\mathcal{R} = \{(u, i, r_{u,i}) : r_{u,i} \neq 0\}$, where $r_{u,i}$ (e.g., 1–5 stars) is user $u$'s rating of item $i$. A CF method learns both user and item *embeddings* from $\mathcal{R}$, where the embedding representation $\mathbf{i}$ of $i \in \mathcal{I}$ captures its latent attributes, and the representation $\mathbf{u}$ of $u \in \mathcal{U}$ reflects their *utility (or preference) function* over these attributes. Suitable CF methods include matrix factorization (Mnih & Salakhutdinov, 2007), or neural CF (Rendle et al., 2020; He et al., 2017; Beutel et al., 2018), whose *two-tower model* processes users and items with separate, but co-trained, networks to produce respective embeddings $\mathbf{u}$ and $\mathbf{i}$.

**Endorsement Task.** A key question when using LMs for endorsing personalized recommendations is how to effectively exploit the information captured by the RS embedding space to generate factual, appealing, and user-tailored endorsements. To measure the LM's efficacy on this task, we develop four metrics: *precision*, *appeal*, *preference coverage*, and *preference prioritization*. These assess the LM's ability to be factual, appeal to users, align with user preferences, and prioritize specific user preferences, which adheres to the *narrative paradigm* (Fisher, 1985), which emphasizes storytelling, coherence, and fidelity as building blocks for meaningful communication. We elaborate on the four criteria below.

*Precision (Prec).*  Recommendation endorsement can be viewed as a form of abstractive summarization (Zhang et al., 2020a; Liu et al., 2022): its text should capture item characteristics that explain why a user would benefit from the item. *Factuality* is critical to RS integrity—an endorsement must describe genuine merits and drawbacks of the item, rather than persuasive distortions or hallucinations.

*Appeal (App).* Increasing attention has been paid to enriching recommendations to appeal to users (Felfernig et al., 2007; Zhang et al., 2020b). To the extent that we do not sacrifice user benefit and well-being, appealing endorsements have value if they encourage users to accept recommendations with significant utility. The *Appeal* of an endorsement depends on various factors including *stylistic variation* (eloquent pitches vs. dry factual summaries) or breadth/depth of explanation, which may be challenging to explicitly quantify.

*Preference Coverage (Pcov).* An endorsement should describe both the positive and negative attributes of an item that the user cares about. This ensures the user can make *appropriate and informed decisions* about the recommended items, and promotes the interpretability and overall integrity of the RS. For example, an endorsement that only articulates positive attributes of an item w.r.t. a user's preferences, while ignoring relevant negative aspects, might be factual and appealing to the user, but could easily mislead the user into making a poor decision. Preference coverage measures the correlation between an endorsement and a user's interpretable preferences. An LM that excels at preference coverage relates recommended items to users by matching facts with system's belief about user preferences.

*Preference Prioritization (Ppr).* A recommendation endorsement should *prioritize communicating aspects of an item that matter most to the user given their preferences*. Preference prioritization focuses on the RS's assessment of the "most important" attributes to the user. This can be captured by the concept of a user's *perceived utility* for an item, i.e., after reading an effective endorsement text, the user's belief about the item's utility should be better than merely reading the full item descriptions. An effective endorsement is one that focuses on attributes that a user cares most about, and conveys how the item matches their preferences. Quantifying preference prioritization is challenging, as it depends on user interpretation of the item endorsements, not merely on their preferences. Finally, though preference prioritization overlaps with coverage, it emphasizes the relevance to a user's preferences.

## 3 Endorsement using Supervised Learning

We first describe a supervised learning approach to solving the RS endorsement task. We assume a dataset $\mathcal{D} = \{(\mathbf{I}^{(k)}, \mathbf{i}^{(k)}, \mathbf{u}^{(k)}, Y^{(k)})\}_{k=1}^{|\mathcal{D}|}$, where each $\mathbf{I}$ is a textual description of item $i \in \mathcal{I}$ (e.g., description, user reviews); $\mathbf{i}$ is the CF embedding of $i$; $\mathbf{u}$ is the CF embedding of a user $u \in \mathcal{U}$; and $Y$ is the endorsement text. (See Appendix C for details on $\mathcal{D}$'s generation.)

Given a set of items, the role of an LM is to predict the probability of response $Y$ given the item and user contexts. A standard way to apply an LM to this task is to use a transformer (Wolf et al., 2019) T to encode an item's text $\mathbf{I}$ as an $N_{\mathbf{I}}$-length sequence of embeddings $(z_0, \ldots, z_{N_{\mathbf{I}}-1})$ induced by T's attention layers, where $N_{\mathbf{I}}$ is an upper-bound on the length (number of tokens) of an item description $\mathbf{I}$. The endorsement text $Y = \{y_n\}_{n=0}^{N-1}$ is sampled token-by-token in an auto-regressive manner using a decoder $\Psi$: $Y \sim \Psi(\cdot \mid z) := \prod_{n=0}^{N-1} \Psi(y_n \mid y_0, \ldots, y_{n-1}; z)$, where $y_0$ is a fixed start-of-sentence token (Chien & Kuo, 2019). However, this architecture assumes all inputs are textual. In order to generate personalized endorsements, we wish to include the item and user CF embeddings as inputs, as they capture important behavioral information.

**Personalizing the LM via User Profile Prompting.** One approach to overcome this limitation is to assume access to a textual *user profile*, which maps a user $u$ (or their CF-embedding $\mathbf{u}$) to a textual preference description $\{U_k(\mathbf{u})\}_{k=1}^K$ (e.g., $K$ bullet-point preference texts that reflect $\mathbf{u}$). One may extend this approach to incorporate personalization by encoding both the item and user texts with the text-only transformer T. However, the quality of this personalized LM hinges on how well user preferences can be represented in natural language. Indeed, crafting effective user summaries is a highly non-trivial task (Radlinski et al., 2022). Instead, we *directly inject* the user CF embeddings into the LM, as described below.

**Injecting Embeddings into the LM.** We directly inject user-item behavioral information into a seq2seq LM (Vaswani et al., 2017) to generate personalized endorsements. Specifically, we augment the standard LM with *adapters* (Pfeiffer et al., 2020) $W_I, W_U : \mathcal{V} \mapsto \mathcal{Z}$, to induce a new LM $\Phi := \Psi \circ (T \times W_I \times W_U)$ (Tennenholtz et al., 2024). Here, T maps text-input tokens to $\mathcal{Z}$, whereas $W_I$ (resp., $W_U$) maps item (resp., user) CF-embedding vectors $\mathcal{V}$ to $\mathcal{Z}$. Importantly, T, $W_I$, and $W_U$ map tokens and CF vectors to a common space so that their relationship can be captured by the transformer's attention mechanism.

**Supervised Training.** Treating the LM $\Phi$ as a factored distribution of item-user information over endorsement tokens, one way to train $\Phi$ is with *behavioral cloning (BC)* (Sasaki & Yamashina, 2020), which maximizes the conditional log-likelihood w.r.t. to dataset $\mathcal{D}$:

$$\Phi_{\text{BC}}(\cdot \mid y_0; \mathbf{I}, \mathbf{i}, \mathbf{u}) \in \arg\min_{\Phi} \ L_{\text{Cond}}(\Phi) := -\mathbb{E}_{(\mathbf{I}, \mathbf{i}, \mathbf{u}, Y) \sim D}\Big[ \sum_{n=0}^{N-1} \log \Phi(y_n \mid y_0, \ldots, y_{n-1}; \mathbf{I}, \mathbf{i}, \mathbf{u}) \Big].$$

Here, $\Phi$ predicts $\mathbb{P}(Y \mid y_0, \mathbf{I}, \mathbf{i}, \mathbf{u})$ of the endorsement $Y$ conditioned on item context $(\mathbf{I}, \mathbf{i})$ and user embedding $\mathbf{u}$. The text-based sub-models within $\Phi$ (encoder T and decoder $\Psi$) can be warm-started with pre-trained LM checkpoints (e.g., PaLM2) which encode rich semantic information. The adapter layers $W_I$ and $W_U$ are trained from scratch.

**Two-Stage Training.** As we use distinct sub-model initialization schemes, the pre-trained LM modules have established embeddings in the language space, while the newly initialized adapters require more training to map CF embeddings to another latent space. Thus, training the full model may cause the LM to disregard the embedding contexts (with diminished adapter weights) and fall back to the pre-trained token generation probabilities. Effectively, $\Phi$ degenerates to a non-contextual LM that ignores user embedding input. To alleviate this, we use the *two-stage* BC procedure of Tennenholtz et al. (2024): we first perform *cold-start* training of the adapters $W_I$ and $W_U$ on an auxiliary task, fixing transformer parameters (T, $\Psi$) as non-trainable; we then train the entire model using the maximum likelihood objective. Alternatively, we can leverage parameter-efficient training approaches (e.g., low-rank adaptation (Hu et al., 2021) for greater training efficiency at the second stage). This two-stage training is critical for LM convergence (Tennenholtz et al., 2024).

# 4 Optimal Endorsement using P⁴LM

While the supervised approach above may generate seemingly good endorsements, it may not adhere to the four principles outlined in Sec. 2. We now develop a rigorous learning framework, namely *aPpealing, Precise, Preference-comprehensive and Prioritized Language Modeling* (P⁴LM), to optimize an endorsement LM using RLAIF. (See Algorithm 1 and Fig. 1 for an overview of P⁴LM.) Recall that our objective is to train an LM to articulate factual and relevant item nuances that align with the user's preferences. Here, the LM acts as a policy which maps text inputs and user/item behavioral embedding vectors to endorsements.

In RLAIF, rewards are computed using feedback from another LM. RLAIF can effectively align LMs with specific metrics, using labels generated by off-the-shelf LMs rather than by human raters (i.e., RL fine-tuning with human feedback (RLHF)). That said, recent work has shown that hybrid human-AI preference models, together with *self-improving* fine-tuning, outperforms traditional supervised fine-tuned baselines and offers additional benefits (Lee et al., 2023; Bai et al., 2022; Zhu et al., 2023). Using the four criteria for LM-based RSs outlined in Sec. 2, we develop four reward models to train and evaluate LMs w.r.t. precision, appeal, prefer-

---

**Algorithm 1** P⁴LM Learning Framework

1: **Input:** Off-the shelf NLI model; pretrained LM.
2: **Data Generation**: Generate supervised data using self-critiquing (App. C) for precision, appeal, preference coverage and prioritization.
3: **Supervised Learning**: Finetune LM using data from previous step and user embeddings (Sec. 3).
4: **Reward Models:** Generate preference labels using LM for appeal, preference coverage, and preference prioritization and train reward models (Sec. 4.1).
5: **RLAIF:** Finetune LM with RL, using the aggregated RM from previous step (Sec. 4.2). Use finetuned supervised model of step 3 as anchor model.

---

ence coverage, and preference prioritization. We then devise an RLAIF technique to fine-tune such LMs with a joint reward model that embodies these four components.

## 4.1 Reward Models

We describe the reward models (RMs) corresponding to each of the criteria defined in Sec. 2.

**Precision (Prec) RM.**[1] Inspired by Roit et al. (2023) and Honovich et al. (2022), we evaluate *factuality* in our LM-based RS using an *entailment reward* (Bowman et al., 2015). Unlike widely-used metrics (e.g., ROUGE (Lin, 2004)), which are ineffective at hallucination detection, we adopt a *textual entailment*, or *natural language inference (NLI)* metric to measure factuality of our generated text, viewing it as a partial summary of an items's description. Specifically, we define the NLI score, i.e., $\text{Prec}(Y; \mathbf{I})$, as the probability of entailment w.r.t. a classifier trained on entailment datasets (e.g., MacCartney & Manning (2007)).

**Appeal (App) RM.** To assess appeal, we use a dataset of pairwise human/machine demonstrations (see Appendix C for details on its construction). We develop an *appeal model* which scores the generated text $Y$ to assess its appeal, using learning from human/AI feedback (LAIF) (Christiano et al., 2017). Let $\mathcal{D}_{\text{app}} = \{(Y_w^{(k)}, Y_l^{(k)}; \mathbf{I})\}_{k=1}^{|\mathcal{D}_{\text{app}}|}$ be the labeled dataset reflecting the relative appeal of two texts $Y_w, Y_l$, where $Y_w$ is more appealing given item description $\mathbf{I}$, i.e., $Y_w \succ Y_l | \mathbf{I}$. Assuming these relationships are governed by a model $\text{App}(Y; \mathbf{I})$, we parameterize it via Bradley-Terry (Huang et al., 2006), giving an appeal distribution: $p_{\text{app}}(Y_w \succ Y_l; \mathbf{I}) = \exp(\text{App}(Y_l; \mathbf{I})) / (\exp(\text{App}(Y_w; \mathbf{I})) + \exp(\text{App}(Y_l; \mathbf{I})))$. By formulating the problem as a binary classification task, the learned latent model minimizes the negative log-likelihood loss: $-\mathbb{E}_{(Y_w, Y_l; \mathbf{I}) \sim \mathcal{D}_{\text{app}}} \log \sigma(\text{App}(Y_w; \mathbf{I}) - \text{App}(Y_l; \mathbf{I}))$. To reduce variance, we subtract this model with its population mean so that $\mathbb{E}_{(Y, \mathbf{I}) \sim \mathcal{D}_{\text{app}}}[\text{App}(Y; \mathbf{I})] = 0$.

---

[1]Here, "Precision" denotes factual articulation of item information, distinct from its common usage in information retrieval or ML.

**Preference Coverage (Pcov) RM.** To explicitly relate item features to user preferences, for each item-user-endorsement tuple $(\mathbf{I}, \mathbf{u}, Y)$, we assume access to a user profile $\{U_k(\mathbf{u})\}_{k=1}^{K}$ (a mapping from a user's CF embedding to $K$ textual preference descriptions). Quantifying preference coverage can then be accomplished by endorsement-preference similarity matching. Specifically, for each tuple, the *preference coverage* score $s$ can be defined as either the *average cosine similarity* of the encoded endorsement $T(Y)$ with $K$ encoded user preference texts $\{T(U_k(\mathbf{u}))\}_{k=1}^{K}$. Alternatively, we use a *black-box estimated likelihood*, e.g., via prompting an LLM with query: "Does endorsement $Y$ sufficiently cover the preferences depicted by user profiles $\{U_k(\mathbf{u})\}_{k=1}^{K}$," and predicting the likelihood of the answer.

We train a scoring model $\text{Pcov}(Y; \mathbf{I}, \mathbf{u})$ by minimizing an $\ell_2$ regression loss $\mathbb{E}_{(\mathbf{I}, \mathbf{u}, Y, s)}(s - \text{Pcov}(Y; \mathbf{I}, \mathbf{u}))^2$. We can also employ LAIF (Christiano et al., 2017) to learn a preference coverage model. Assuming the endorsement text $Y$ covers more preference attributes in the user profile than item descriptions themselves (i.e., $Y \succ \mathbf{I}|\mathbf{u}$), the model $\text{Pcov}(Y; \mathbf{I}, \mathbf{u})$ can be learned by minimizing the regularized negative log-likelihood: $-\mathbb{E}_{(Y, \mathbf{I}, \mathbf{u})} \log \sigma(\text{Pcov}(Y; \mathbf{I}, \mathbf{u}) - \text{Pcov}(\mathbf{I}; \mathbf{I}, \mathbf{u})) + \lambda \cdot \mathbb{E}_{(\mathbf{I}, \mathbf{u}, Y, s)}(s - \text{Pcov}(Y; \mathbf{I}, \mathbf{u}))^2$, for $\lambda > 0$, which also enforces the equality constraint $s = \text{Pcov}(Y; \mathbf{I}, \mathbf{u})$.

**Preference Prioritization (Ppr) RM.** We define a scoring model $\text{Ppr}(Y; \mathbf{i}, \mathbf{u})$ to quantify perceived utility, or how an endorsement connects an item with the most important preferences of the user. One special case is when the endorsement is a complete item description, i.e., $Y = \mathbf{I}$. In this case, we may assume that Ppr is simply the item utility, i.e., $\text{Ppr}(\mathbf{I}; \mathbf{i}, \mathbf{u}) = \mathbf{i} \cdot \mathbf{u}$. Alternatively, we can employ LAIF (Christiano et al., 2017), leveraging pairwise feedback to learn Ppr. Using the dataset $\mathcal{D}$ of item descriptions $\mathbf{I}$, user-item CF embeddings $\mathbf{u}, \mathbf{i}$, and endorsement texts $Y$, and assuming this text has greater preference prioritization than item descriptions (i.e., $Y \succ \mathbf{I}|\mathbf{i}, \mathbf{u})^2$ a Bradley-Terry model $\text{Ppr}(Y; \mathbf{i}, \mathbf{u})$ is learned by minimizing the negative log-likelihood: $-\mathbb{E}_{(Y, \mathbf{I}; \mathbf{i}, \mathbf{u})} \log \sigma(\text{Ppr}(Y; \mathbf{i}, \mathbf{u}) - \text{Ppr}(\mathbf{I}; \mathbf{i}, \mathbf{u}))$. This loss can be further regularized with the term $\lambda \cdot \mathbb{E}_{(\mathbf{I}; \mathbf{i}, \mathbf{u})}(\text{Ppr}(\mathbf{I}; \mathbf{i}, \mathbf{u}) - \mathbf{i} \cdot \mathbf{u})^2$, for $\lambda > 0$, which enforces the constraint: $\text{Ppr}(\mathbf{I}; \mathbf{i}, \mathbf{u}) = \mathbf{i} \cdot \mathbf{u}$.

### 4.2 Reward Aggregation and RL-based Fine-tuning

In multi-objective RL, reward models can often be aggregated using *linear scalarization* (Peschl et al., 2021), which solves for an optimum on the convex Pareto frontier. Given an endorsement text $Y$, item description $\mathbf{I}$, and user-item embeddings $(\mathbf{u}, \mathbf{i})$, we define the multi-objective reward as:

$$r(Y; \mathbf{I}, \mathbf{i}, \mathbf{u}) = \begin{cases} \eta_1 \text{Prec}(Y; \mathbf{I}) + \eta_2 \text{App}(Y; \mathbf{I}) + \eta_3 \text{Ppr}(Y; \mathbf{i}, \mathbf{u}) + \eta_4 \text{Pcov}(Y; \mathbf{I}, \mathbf{u}) & y_n = [\text{EOS}]; \\ 0 & \text{o.w.,} \end{cases} \tag{1}$$

where $\eta_1, \eta_2, \eta_3, \eta_4 \geq 0$ are the importance weights for the component rewards.

Wang et al. (2024) show that, when the reward models are either pointwise logit or pairwise logit-difference models (as in our case), one can simply employ the reward aggregation technique with all importance weights set to 1, without losing optimality. To show this, Wang et al. (2024) first apply a log-sigmoid transformation to all reward models. In our case, the point-wise Prec reward (learned with logistic regression) becomes $\log \sigma(\text{Prec}(Y; \mathbf{I}))$, and corresponds to the log-probability of the positive entailment, i.e., $\log \mathbb{P}(\text{Entail} = 1)$, where Entail is the entailment random variable. Similarly, for the pairwise rewards that characterize appeal, preference coverage, and preference prioritization (trained to maximize logit differences between a pair of endorsement scores), the log difference between the score of endorsement $Y$ and its baseline counterpart can be written as $\log \sigma(R(Y; \mathbf{I}, \mathbf{i}, \mathbf{u}) - R(Y_{\text{ref}}; \mathbf{I}, \mathbf{i}, \mathbf{u}))$, for $R \in \{\text{App}, \text{Ppr}, \text{Pcov}\}$. This quantity corresponds to the log-likelihood of positive preference, i.e., $\log \mathbb{P}(R_{\text{Pref}} = 1)$, where $R_{\text{Pref}}$ is a random variable which indicates endorsement $Y$ is preferred over the reference text $Y_{\text{ref}}$[3] under the criterion of interest.

---

[2] Instead of comparing the endorsement text with item description, one could construct a dataset with two texts and a labeled rating order (see Appendix C for details).

[3] Response $Y_{\text{ref}}$ can either be the label in offline data $\mathcal{D}$ or the output of a pre-trained model.

Finally, assuming conditional independence of random variables (Entail, $\text{App}_{\text{Pref}}$, $\text{Ppr}_{\text{Pref}}$, $\text{Pcov}_{\text{Pref}}$), adding the transformed rewards is equivalent to a logical-AND operation, i.e.,

$$r(Y; \mathbf{I}, \mathbf{i}, \mathbf{u}) := \log \sigma \text{Prec}(Y; \mathbf{I}) + \sum_{R \in \{\text{App}, \text{Ppr}, \text{Pcov}\}} \log \sigma(R(Y; \mathbf{I}, \mathbf{i}, \mathbf{u}) - R(Y_{\text{ref}}; \mathbf{I}, \mathbf{i}, \mathbf{u})) \tag{2}$$
$$= \log \mathbb{P}(\text{Entail} = 1, \text{App}_{\text{Pref}} = 1, \text{Ppr}_{\text{Pref}} = 1, \text{Pcov}_{\text{Pref}} = 1),$$

which implies that the joint reward aligns across all aspects. This, in turn, shows that all importance weights $\eta_1, \eta_2, \eta_3, \eta_4$ can indeed be set to 1.

Given the LM $\Phi(Y \mid \mathbf{I}, \mathbf{i}, \mathbf{u})$ and joint reward model $r(Y, \mathbf{I}, \mathbf{i}, \mathbf{u})$, the goal of LM fine-tuning is to maximize the overall quality of the generated text: $\max_\Phi \ \mathbb{E}_{(\mathbf{I}, \mathbf{i}, \mathbf{u})} \mathbb{E}_{\Phi(Y|\mathbf{I}, \mathbf{i}, \mathbf{u})}[r(Y; \mathbf{I}, \mathbf{i}, \mathbf{u})]$. Using the MDP framework (Puterman, 2014), this problem can be solved using on-policy RL, e.g., REINFORCE (Williams, 1992), with trajectories generated by the current LM.

One risk of RLAIF fine-tuning is overfitting the LM to the reward model, thereby degrading the LM's inherent "semantic skill." To alleviate this, we add a KL regularization term (Ouyang et al., 2022; Stiennon et al., 2020) to the MDP objective, comparing the LM $\Phi(Y|\mathbf{I}, \mathbf{i}, \mathbf{u})$ and the supervised fine-tuned model $\Phi_{\text{BC}}(Y|\mathbf{I}, \mathbf{i}, \mathbf{u})$. Leveraging the autoregressive nature of LMs, KL regularization is applied over the entire MDP trajectory, reducing the objective to

$$\Phi_{\text{P}^4\text{LM}} \in \arg\max_\Phi \ J(\Phi) := \mathbb{E}_{(\mathbf{I}, \mathbf{i}, \mathbf{u})} \mathbb{E}_{\Phi(Y|\mathbf{I}, \mathbf{i}, \mathbf{u})} \left[ r(Y; \mathbf{I}, \mathbf{i}, \mathbf{u}) - \beta \log \frac{\Phi(Y|\mathbf{I}, \mathbf{i}, \mathbf{u})}{\Phi_{\text{BC}}(Y|\mathbf{I}, \mathbf{i}, \mathbf{u})} \right],$$

corresponding to a KL-regularized MDP, which can be solved by policy gradient methods.

## 5 Experiments

We conduct empirical evaluation of $\text{P}^4\text{LM}$, assessing whether our reward models can significantly increase the precision, appeal, preference coverage, and preference prioritization of item endorsements. We conduct experiments on two recommendation datasets: (i) MovieLens 25M (Harper & Konstan, 2015), which contains ratings of $62,423$ movies by $162,541$ users, and (ii) Amazon Product Review (Ni et al., 2019), from which we use the "Clothing, Shoes and Jewelry" category, containing 5.7M ratings of 1.5M products. We use user-item interactions to generate item descriptions, user-preference texts, and recommendation endorsements by prompting a PaLM2-L LM (Google et al., 2023) (see Appendix C for details). The resulting datasets have four components: (1) item descriptions $\mathbf{I}$, (2) user-item behavioral embeddings $(\mathbf{i}, \mathbf{u})$, (3) user preference texts $\text{U}(\mathbf{u})$, and (4) endorsements $Y$.

We experiment with PaLM2-XS (Google et al., 2023) and incorporate user and item embeddings (see Sec. 3) by augmenting the model with adapter layers. To simplify hyper-parameter optimization, we use reward aggregation (Equation 2) to combine the rewards. We also experiment with rewards combined with various importance weights (see Appendix A).

To demonstrate the efficacy of $\text{P}^4\text{LM}$, we compare it with the following SOTA baselines: (i) **PaLM2-L**, a pre-trained text-only large LM, prompted using item descriptions, user preference texts and instructions to generate a response that adheres to our four endorsement principles; (ii) **Supervised Fine-Tuned with Text (SFT-Text)**, a PaLM2-XS model fine-tuned with the datasets above, with explicit user-item texts as input; (iii) **Supervised Fine-Tuned (SFT)**, a PaLM2-XS model fine-tuned to use user-item embedding vectors.[4] We measure performance of all methods using both *model-based* and *human* evaluation.

**Model-Based Evaluation.** We conduct *model-based* (MB) evaluations on both the movie and product endorsement tasks using our four RMs from Section 4.1, i.e., Prec, App, Ppr and Pcov.

---

[4]Text-based approaches are inherently limited by the specificity of textual user profiles. Defining optimal textual profiles is beyond this work's scope. On the other hand, by directly injecting user embeddings into the word-token embedding space, $\text{P}^4\text{LM}$ can leverage diverse user data, reducing reliance on specific prompts. The output from adapter $W_{\mathcal{U}}$ efficiently represents user details, minimizing dependence on profile formatting.

Table 1: Evaluation of MB metrics on Amazon dataset: Prec scores from NLI model entailment probabilities; average logits for other metrics from three RMs.

| Method | Prec | Ppr | App | Pcov |
|---|---|---|---|---|
| | | Evaluation Metrics | | |
| PaLM2-L | 0.34 | –1.73 | –5.47 | –5.48 |
| SFT-Text | 0.37 | –1.70 | –5.48 | –5.42 |
| SFT | 0.37 | –1.59 | –5.54 | –5.33 |
| $P^4$LM (ours) | **0.79** | **–1.47** | **–5.27** | **–4.93** |

Table 2: Human evaluation: Model vs. SFT win rates, covering Prec, Ppr, App Ratios, Pcov increase, and overall quality (see Appendix E).

| | Method | Prec | Ppr | App | Pcov | All |
|---|---|---|---|---|---|---|
| | | | Evaluation Metrics | | | |
| MLens | PaLM2-L | 0.31 | 0.35 | 0.31 | 8.3% | 0.33 |
| | SFT-Text | 0.5 | 0.5 | 0.50 | 2.1% | 0.49 |
| | $P^4$LM (ours) | **0.77** | **0.64** | **0.69** | **35%** | **0.71** |
| Amazon | PaLM2-L | 0.43 | 0.40 | 0.49 | –17% | 0.47 |
| | SFT-Text | 0.52 | 0.27 | 0.37 | –32% | 0.44 |
| | $P^4$LM (ours) | **0.73** | **0.67** | **0.64** | **17%** | **0.81** |

We report the scores of endorsements $Y$ generated by each model on a held-out, unlabeled dataset $\mathcal{D}_{\text{test}}$ consisting of 73 (Amazon) or 96 (MovieLens) non-overlapping user-item pairs. The average results are presented in the main text, with detailed results and confidence intervals available in Appendix A. We also assess the relative improvement of each LM over the standard SFT baseline by computing (a) the *win rate* (number of occurrences on which the tested LM outperforms SFT), and (b) the *absolute increase* (of LM's score improvement).

To validate the efficacy of our four reward models, in reflecting corresponding preference values in the context of recommendation endorsement, we first assess their accuracy. We start by prompting the Gemini Ultra LLM (Google, 2023) (see Appendix A.1 for details) to generate a synthetic test dataset of endorsement pairs with clear preferences on precision, appeal, preference coverage and prioritization respectively. Then the RMs are evaluated based on their abilities to distinguish the endorsements with respect to the corresponding principles. The results presented in Table 3 (Appendix A) illustrate that the RMs can effectively classify all the test cases, thus certifying the validity of these models.

Table 1 depicts the MB evaluation results based on the four principles of endorsements on the Amazon dataset. It highlights the robust performance of $P^4$LM in all four pivotal dimensions: precision, appeal, preference coverage and prioritization. $P^4$LM attains the highest precision score by a large margin, underscoring its ability to mitigate the risk of hallucinating information about recommended items. It also fares well on all other categories, corroborating the hypothesis that RL-based fine-tuning, when paired with reward aggregation, manages to align the endorsement LM with respect to all the reward models. Similar evaluation on the MovieLens dataset can be found in Table 5 (Appendix A). Though similar trends (especially in the precision category) are observed, PaLM2-L achieves the best performance in appeal, preference coverage and prioritization. This is unsurprising because (i) PaLM2-L is much larger than $P^4$LM; (ii) Different from the specialized user-item information associated with Amazon products, there is ample public data about movie contexts and generic user preferences available for

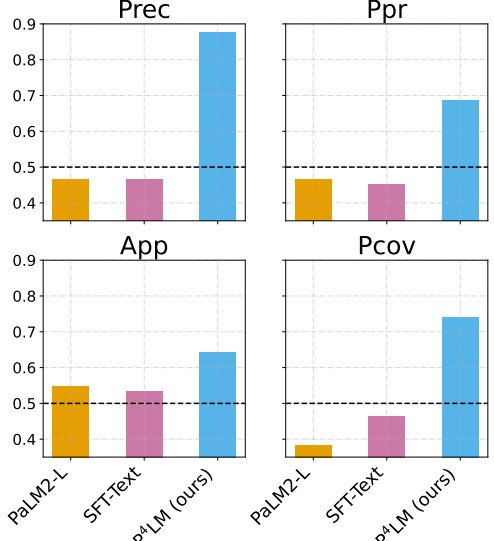

Figure 2: Win rates of MB scores against SFT from the Amazon dataset. Dotted lines at 0.5 represent ties. The win rate, as defined in Appendix B, is calculated by comparing the MB score from output against the SFT score.

pre-training such that PaLM2-L can adeptly generate personalized endorsements. These observations are also correlated with the findings highlighted in Figures 3 and 5 (both in Appendix A) for MovieLens, Figures 2 and 4 (Appendix A) for Amazon, which elucidate

both the win rates and absolute score increases of each method when compared with the SFT baseline.

Comparing SFT and SFT-Text, both methods exhibit comparable performance across most metrics, except for the preference coverage and prioritization scores. SFT extracts user preferences from the behavioral embedding to generate personalized endorsements, while SFT-Text generate outputs that attends to specific user preferences in the input prompt. However, the dependency on text inputs introduces limitations, omitting subtle user preferences captured by their behavioral embedding. This difference suggests that text descriptions are unable to fully reflect the intricate user preference information in embeddings.

**Human Evaluation.** Table 2 shows the human evaluation on models trained with Movie-Lens and Amazon datasets, in terms of their relative preferences over the SFT baseline by human *raters*. We asked raters to compare two endorsement texts, one from SFT and another from a test model, to assess relative performances. Raters evaluated these endorsements using the same four criteria as in the model-based evaluation, with an additional criterion for overall quality comparison (see Appendix E for details).

Raters assess the endorsements generated by different models, including P$^4$LM. For the LLMs trained with Amazon data, P$^4$LM unanimously outperforms other baselines across all the four endorsement principles, in most cases with significant margins. In particular, human evaluation much favored the precision and preference coverage of P$^4$LM, both consistent with the model-based observations. Interestingly, in the MovieLens experiment human evaluation highlighted P$^4$LM's superior performances across all the metrics (see Table 2), while model-based evaluation only indicated the same trend on factuality (see Appendix A Table 5). These discrepancies may be attributed to the approximation errors of the reward models (Ppr App, Pcov), which are sensitive to their preference training data. Nevertheless, raters showed a strong preference for P$^4$LM in terms of overall text quality.

Human evaluation also favored SFT over SFT-Text in the Amazon domain, but is indifferent in the MovieLens domain. This may likely due to SFT-Text, which shares inputs with PaLM2-L, is also trained with its generated dataset and thus limited by PaLM2-L's semantic skills. Contrarily, SFT, while trained with the same data, can also leverage additional information from user's behavioral embeddings to enhance the LM's understanding about user's preferences, giving it an edge over SFT-Text, especially in preference coverage and prioritization scores. This observation is more pronounced in the Amazon experiment, of which the user-item information is less available in the public data.

**Ablation Studies** Our ablation studies (Table 7 for MovieLens, Table 8 for Amazon) show that P$^4$LM trained using RLAIF with a single RM scores higher in MB evaluation for the corresponding RM of interest. Intriguingly, in both domains, models trained solely on preference prioritization or preference coverage excel on both metrics, suggesting correlation between these two categories, which is somewhat expected (see Sec. 2). Human evaluations on models trained with individual RMs are detailed in Appendix A Table 9. Notably, ranking of these models by raters differs from the MB ablation. For example, on MovieLens the Pcov-model (which targets preference coverage) and on Amazon the Ppr-model (which targets preference prioritization), surprisingly score highest on all categories in human evaluations. Human raters are somewhat biased towards endorsements that relate item features to user preferences, heavily influencing their judgment of overall quality.

## 6   Related Work

Our work connects research from personalized RSs, LMs, RL, recommendation integrity.

**Personalized RSs.** RSs permeate e-commerce, content systems, social media, etc. CF methods still play a prominent role (Schafer et al., 2007; Mnih & Salakhutdinov, 2007), with deep learning methods (e.g., neural CF (He et al., 2017), dual encoders (Yi et al., 2019; Yang et al., 2020)) extending traditional CF methods. RSs are increasingly modeling more

nuanced and often sequential aspects of user behavior (van den Oord et al., 2013; Covington et al., 2016; Gauci et al., 2018; Ie et al., 2019; Chen et al., 2019a).

**Conversational RSs & LMs.** Conversational RSs augment traditional RSs with a conversational agent that supports user interaction through natural language dialogue (Chen et al., 2019b; Zhou et al., 2020; Lei et al., 2020; Li et al., 2018; Sun & Zhang, 2018; Christakopoulou et al., 2016). Such systems provide a richer means of understanding a user's preferences, allowing the natural refinement of recommendations and more user control. Conversational RSs are a ready target for the use of LMs, though leveraging LMs in RSs is a relatively recent effort. With the advance of transformer architectures (Vaswani et al., 2017; Wolf et al., 2019), LMs have found use cases beyond typical NLP tasks, including the synthesis of textual data with user preferences to enhance RS personalization and expressiveness (Jaech & Ostendorf, 2018; Xia et al., 2023). Our work fits into this space, aiming to generate appealing narratives that effectively communicate a recommendation's coverage to the user.

**Transparency and Factuality in RSs.** Maintaining integrity in RSs is vital, but challenging, given the potential for RSs to inadvertently mislead users or reinforce biases (Abdollahpouri et al., 2019; Shen et al., 2023; Cabello et al., 2023). Increasingly, research has delved into the fairness, transparency, and interpretability of RS algorithms (Beutel et al., 2019; Ghazimatin et al., 2020; Chen et al., 2023). Our work emphasizes factual and precise recommendations that articulate genuine merits rather than distortions designed to appeal to users.

**RLHF & RLAIF.** The integration of RL with LMs has emerged as an appealing strategy for refining models beyond supervised fine-tuning (Williams, 1992; Ranzato et al., 2016). The RLHF methodology (Christiano et al., 2017; Bai et al., 2022) allows model responses to be first rated by human evaluators, and then used to fine-tune models (with RL). Inverse RL (Abbeel & Ng, 2004) has been used to acquire objectives from expert demonstrations in textual settings (Daniels-Koch & Freedman, 2022; Sun, 2023). Finally, there is growing interest in AI-based feedback, where responses are labeled by models rather than human raters (Lee et al., 2023; Bai et al., 2022). Our work collectively shows tremendous potential for using RL to drive LMs for better alignment with user preferences and RS objectives.

## 7  Conclusion

We have studied language modeling for generating recommendation endorsements, proposing novel reward models which quantify critical aspects of effective personalization, and training our LM using RLAIF with these rewards. The resulting LM, $P^4LM$, not only supports language interaction with a user, but understands latent user preferences as encoded in a CF embedding space. It provides item endorsements that are factual, appealing, preference comprehensive and prioritized, connecting relevant item attributes with a user's preferences, and increasing the odds of users accepting high-value recommendations. Experiments on benchmark datasets demonstrated the efficacy of $P^4LM$ across all these dimensions.

Our work is a step toward creating a robust conversational RS that can identify the intricate (and most relevant) connections between an item's features and a user's preferences and explain them in an appealing fashion. Future work includes: improving $P^4LM$'s ability to generate longer responses, e.g., by moving beyond single-shot, autoregressive decoding; extending RL fine-tuning to handle multi-turn conversational recommendations; developing better reasoning capabilities that trade off user-item preferences and constraints; and expanding LM's functionality beyond recommendation, to include technical support, negotiation, etc.

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

# A    Additional Results

Table 3: The accuracies of the RMs on the test set generated with Gemini Ultra Google (2023).

|  | Prec | Ppr | App | Pcov |
|---|---|---|---|---|
| MovieLens | 0.875 | 1.0 | 1.0 | 1.0 |
| Amazon | 0.9686 | 1.0 | 1.0 | 1.0 |

## A.1    Validation of Reward Models

In the development of $P^4LM$, we employed four distinct Reward Models (RMs). The efficacy of these RMs is critical, as they need to accurately reflect human values and preferences in the context of endorsement text generation.

The core objective of the evaluation in this part is to determine whether the RMs can reliably distinguish between varying qualities of endorsement texts in terms of factual consistency, appeal, preference coverage, and preference prioritization, respectively. This involves assessing if a specific RM, such as the Prec RM for factual consistency or the Appeal RM for textual appeal, can correctly score texts in alignment with their actual quality.

To validate the effectiveness of these RMs, we generated synthetic examples using Gemini Ultra Google (2023). Our approach involved arbitrarily selecting 32 pairs of movies and users. For each pair, Gemini Ultra was prompted to generate two endorsement texts. These texts were designed to have distinctly different qualities in terms of a specific measure (e.g., one text being more appealing or covering more user preferences than the other, or one being more factually consistent). See the previous page for the prompts we used for the generation of these texts.

The evaluation criteria were straightforward: for each pair of texts, we already knew which one was superior based on the targeted measure. The task for the RMs was to correctly classify these texts — identifying the one with higher quality in the context of the specific measure being evaluated. The accuracy of the RMs in correctly classifying the superior text was calculated, providing a clear metric of their effectiveness. The outcomes of this evaluation are presented in Table 3 for both the MovieLens and the Amazon datasets.

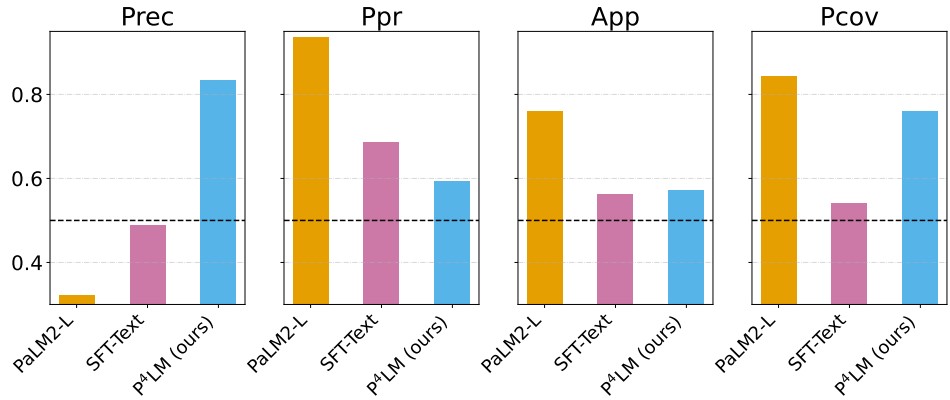

Figure 3: Win rates of model-based scores vs. SFT (MovieLens)

## A.2    Model-based Evaluation

We extend our MB evaluation to the MovieLens dataset, focusing on the performance of $P^4LM$ and other models across the four reward metrics. Table 5 provides these details,

Table 4: MB Evaluation Results (Amazon)

| Method | Evaluation Metrics | | | |
|---|---|---|---|---|
| | Prec | Ppr | App | Pcov |
| PaLM2-L | $0.34 \pm 0.04$ | $-1.73 \pm 0.04$ | $-5.47 \pm 0.07$ | $-5.48 \pm 0.06$ |
| SFT-Text | $0.37 \pm 0.04$ | $-1.70 \pm 0.04$ | $-5.48 \pm 0.08$ | $-5.42 \pm 0.06$ |
| SFT | $0.37 \pm 0.04$ | $-1.59 \pm 0.05$ | $-5.54 \pm 0.07$ | $-5.33 \pm 0.07$ |
| $P^4$LM | $\mathbf{0.79 \pm 0.03}$ | $\mathbf{-1.47 \pm 0.05}$ | $\mathbf{-5.27 \pm 0.04}$ | $\mathbf{-4.93 \pm 0.05}$ |

Table 5: MB Evaluation Results (MovieLens)

| Method | Evaluation Metrics | | | |
|---|---|---|---|---|
| | Prec | Ppr | App | Pcov |
| PaLM2-L | $0.14 \pm 0.02$ | $\mathbf{-1.80 \pm 0.03}$ | $\mathbf{-1.43 \pm 0.04}$ | $\mathbf{-2.03 \pm 0.03}$ |
| SFT-Text | $0.28 \pm 0.03$ | $-2.07 \pm 0.02$ | $-1.68 \pm 0.04$ | $-2.36 \pm 0.03$ |
| SFT | $0.29 \pm 0.02$ | $-2.11 \pm 0.02$ | $-1.65 \pm 0.03$ | $-2.34 \pm 0.03$ |
| $P^4$LM | $\mathbf{0.58 \pm 0.02}$ | $-2.14 \pm 0.02$ | $-1.61 \pm 0.03$ | $-2.16 \pm 0.03$ |

Table 6: Single RM Model-based Ablations (Amazon)

| RM | Evaluation Metrics | | | |
|---|---|---|---|---|
| | Prec | Ppr | App | Pcov |
| Prec | $\mathbf{0.84 \pm 0.03}$ | $-1.74 \pm 0.03$ | $-5.65 \pm 0.05$ | $-5.59 \pm 0.05$ |
| Ppr | $0.29 \pm 0.03$ | $\mathbf{-0.93 \pm 0.04}$ | $\mathbf{-4.96 \pm 0.03}$ | $\mathbf{-4.79 \pm 0.05}$ |
| App | $0.36 \pm 0.03$ | $-1.38 \pm 0.05$ | $\mathbf{-4.95 \pm 0.04}$ | $-5.01 \pm 0.06$ |
| Pcov | $0.47 \pm 0.02$ | $-1.37 \pm 0.02$ | $-5.39 \pm 0.02$ | $-4.97 \pm 0.03$ |

Table 7: Single RM Model-based Ablations (MovieLens)

| RM | Evaluation Metrics | | | |
|---|---|---|---|---|
| | Prec | Ppr | App | Pcov |
| Prec | $\mathbf{0.87 \pm 0.01}$ | $-2.21 \pm 0.01$ | $-1.95 \pm 0.03$ | $-2.30 \pm 0.04$ |
| Ppr | $0.41 \pm 0.03$ | $-2.08 \pm 0.02$ | $-1.80 \pm 0.04$ | $-2.15 \pm 0.03$ |
| App | $0.32 \pm 0.02$ | $-2.19 \pm 0.02$ | $\mathbf{-1.57 \pm 0.03}$ | $-2.33 \pm 0.03$ |
| Pcov | $0.37 \pm 0.02$ | $\mathbf{-2.04 \pm 0.02}$ | $-1.95 \pm 0.02$ | $\mathbf{-1.88 \pm 0.03}$ |

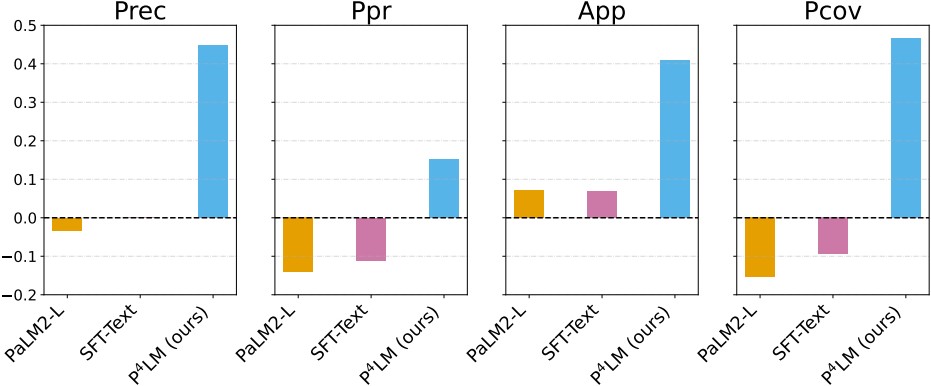

Figure 4: The absolute MB score increases vs. SFT (Amazon)

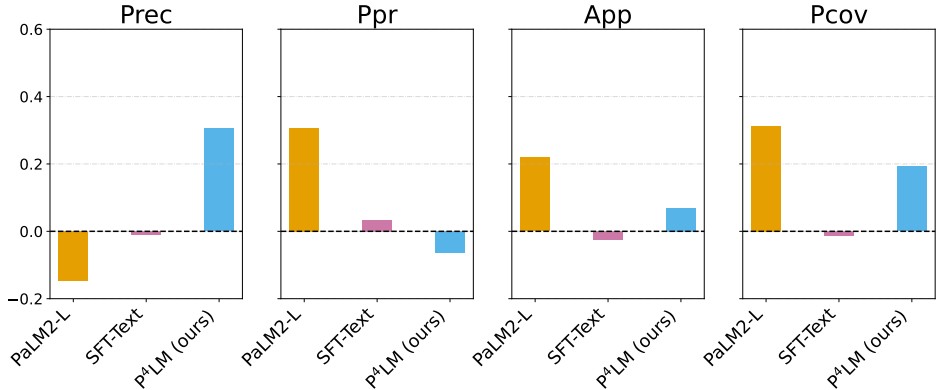

Figure 5: The absolute MB score increases vs. SFT (MovieLens)

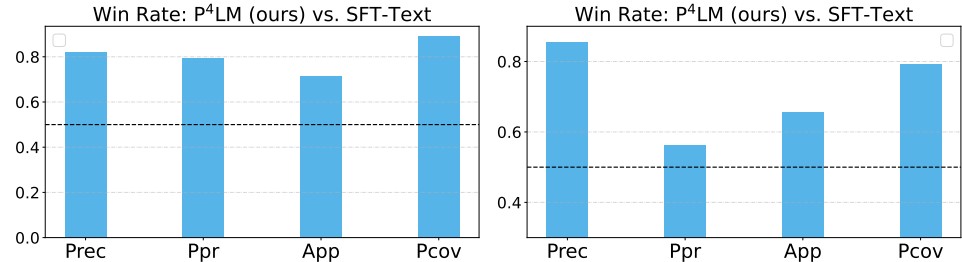

Figure 6: Win rate of P⁴LM over SFT-Text: Amazon (Left) & MovieLens (Right)

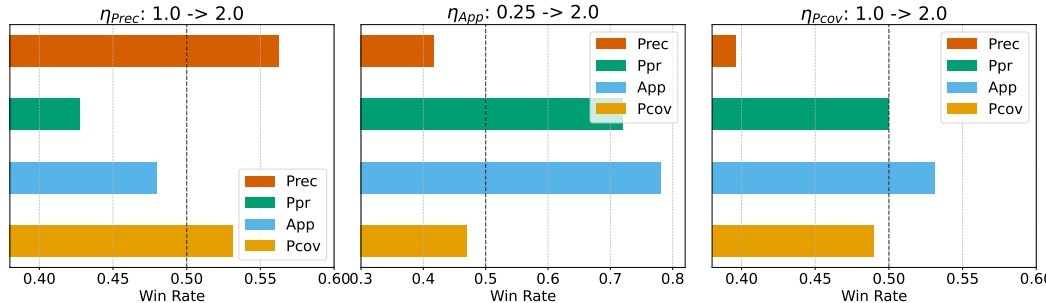

Figure 7: Win Rates While Changing the Mixing Weights of Reward Models

previously summarized in the experimental section (Section 5). While P⁴LM showcases notable precision, it is PaLM2-L that lead in App, Ppr, and Pcov, benefiting from its extensive dataset and larger model size. Interestingly, human evaluations displayed a strong preference for P⁴LM over others, including PaLM2-L, especially in the aspects of precision and preference coverage, suggesting that P⁴LM's endorsements resonated more with human raters despite the mixed results in MB evaluations.

The absolute increases and win rates offer a deeper look into model performances beyond mere score comparisons. While win rates are derived from direct comparisons on individual examples, absolute increases provide an averaged perspective across the test dataset. These metrics (see Appendix B for the precise definitions), detailed in Figures 4 and 5, align with the overall trends noted in our results section.

Comparing P⁴LM to SFT-Text via win rates (Figure 6) reveals that P⁴LM consistently surpasses SFT-Text across all rewards within both the Amazon and MovieLens datasets. Notably, in Preference Prioritization, P⁴LM's enhanced performance underscores its superior

Table 8: Evaluation of MB metrics in the Amazon dataset via individual RMs. Scores, such as Prec in the first row, result from RL fine-tuning an LM with a single RM, then assessing across the four metrics.

| RM | Evaluation Metrics | | | |
|---|---|---|---|---|
| | Prec | Ppr | App | Pcov |
| Prec | **0.84** | –1.74 | –5.65 | –5.59 |
| Ppr | 0.29 | **–0.93** | **–4.96** | **–4.79** |
| App | 0.36 | –1.38 | **–4.95** | –5.01 |
| Pcov | 0.47 | –1.37 | –5.39 | –4.97 |

Table 9: Human evaluation: Single RM-trained Model vs. SFT output win rates. See Appendix E for metric definitions.

| | RM | Evaluation Metrics | | | |
|---|---|---|---|---|---|
| | | Prec | Ppr | App | Pcov |
| MLens | Prec | 0.64 | 0.55 | 0.57 | 6.0% |
| | Ppr | 0.75 | 0.68 | 0.75 | 20% |
| | App | 0.52 | 0.60 | 0.5 | 1.9% |
| | Pcov | 0.81 | 0.72 | 0.80 | 143% |
| Amazon | Prec | 0.64 | 0.61 | 0.62 | 35% |
| | Ppr | 0.82 | 0.78 | 0.84 | 118% |
| | App | 0.70 | 0.52 | 0.66 | 0.0% |
| | Pcov | 0.64 | 0.64 | 0.63 | 29% |

capability for user preference prioritization over SFT-Text, despite SFT-Text's higher win rate relative to SFT. P[4]LM's high win rates across all the metrics also highlight its proficiency in effectively utilizing RS embeddings, a contrast to text-based methods, allowing for a more nuanced capture and representation of user preferences. This distinction serves to spotlight the potential of embedding-centric approaches in advancing personalized recommendation generation.

### A.3 Ablations

Our ablation studies on the MovieLens domain, detailed in Table 7, examine the model's performance when trained exclusively with individual RMs. This approach mirrors our analyses for the Amazon dataset, as discussed in Section 5 (Table 8). Training with singular RMs allows us to isolate and understand the impact of each reward on the endorsement quality across different metrics. As alluded to, models trained with a focus on preference prioritization or preference coverage not only excelled in their respective metrics but also showed commendable performance in other areas. This outcome highlights the intricate relationships between different aspects of endorsement quality, suggesting that improvements in one dimension can positively influence others.

Figure 7 shows the win rate of policies trained with specific reward mixing weights, compared to a base policy trained with another reward mixing weights. The left figure illustrates the scenario when the Prec reward weight is increased from 1.0 to 2.0. Here, we computed the ratio of instances where the policy with an Prec weight of 2.0 achieved higher RM scores than the one with 1.0 by analyzing 96 pairs of (movie, user) and their corresponding endorsement texts. This win rate was calculated for all four RMs: Prec, Ppr, App, and Pcov, offering a nuanced view of how altering a single RM weight influences the overall policy performance across different metrics.

Further exploration into the impact of varying the mixing-weight combinations, presented in Figure 7, reveals several noteworthy trends:

1. Increasing the focus on App positively influences Ppr, suggesting a symbiotic relationship between these two metrics.
2. Enhancing certain metrics often results in a decrease in others, highlighting the inherent trade-offs in our multi-objective optimization problem.
3. Emphasizing Prec slightly increases the Pcov score, though not significantly.

Despite these complexities, our main experiments leverage a reward aggregation technique Wang et al. (2024) to simplify the optimization process, sidestepping the challenges of manually tuning reward mixing weights. This ablation study, therefore, offers additional insights into the potential interdependencies between different RMs, enriching our understanding of how to best align model training with multiple endorsement criteria.

## A.4 Example Outputs

In this part, we conduct a comparative analysis of example decoded output texts derived from SFT and P⁴LM to qualitatively understand their performances and the behaviors learned. Specifically, our focus is on instances where one or more scores attributed to P⁴LM are significantly higher than those derived from SFT. This approach is aimed at determining whether the scores effectively capture the correct information from the generated endorsements.

**Example 1:** (P⁴LM achieving a much higher Precision score than SFT)

---

**Movie: Up (2009)**

Up (2009) is an animated adventure comedy-drama film released by Pixar. The film centers on a grumpy widower named Carl Fredricksen, who, having finally had his fill of living in modern society, ties thousands of balloons to his house and sets off to fulfill his childhood dream of exploring the wilds of South America. However, he soon finds that he is not as alone on his journey as he had originally thought. Up explores the themes of childhood, growing old, and the importance of following one's dreams.

---

**User Preferences**

* They like action movies.
* They like movies with a lot of special effects.
* They like movies with a dark and gritty tone.
* They like movies with a lot of violence.
* They like movies with a lot of suspense.
* They don't like animated movies.
* They don't like romantic comedies.
* They don't like movies with a lot of dialogue.
* They don't like movies with a slow pace.
* They don't like movies with a lot of philosophical themes.

---

**SFT**

Up is a great choice for viewers like you. It is an action-packed adventure comedy-drama with a strong male lead, stunning visuals, and an inspiring story about following one's dreams. It also explores the themes of childhood, growing old, and the importance of family. I highly recommend it!

---

**P⁴LM**

Up (2009) is a Pixar animated adventure comedy-drama film that follows a grumpy widower who sets off on a journey to fulfill his childhood dream of exploring South America. The film is visually stunning and features strong performances from its voice cast. It's also thought-provoking and emotionally resonant, exploring the themes of childhood, growing old, and the importance of following one's dreams. Up is a must-watch for fans of animation, Pixar, and coming-of-age stories.

---

In this example, P⁴LM strives to replicate and paraphrase the content from the provided movie synopsis to uphold high precision. Conversely, SFT crafts an endorsement while unfortunately incorporating misinformation, as indicated in red.

**Example 2:** (P⁴LM achieving a much higher Appeal score than SFT)

---

**Movie: Sherlock Holmes: A Game of Shadows (2011)**

Sherlock Holmes (Robert Downey Jr.) and Dr. Watson (Jude Law) are on the trail of criminal mastermind Professor Moriarty (Jared Harris) in this sequel to the 2009 film "Sherlock Holmes." As the duo follows a series of clues across Europe, they discover that Moriarty is planning a terrorist attack that could change the course of history. The movie is full of action, suspense, and humor, and features strong performances from its leads.

---

**User Preferences**

* They enjoy movies with a lot of action and adventure.
* They like movies that are visually stunning.
* They appreciate movies with a strong story line.
* They enjoy movies that are set in fantasy worlds.
* They like movies that are dark and gritty.
* They dislike movies that are slow-paced.
* They dislike movies that are too predictable.
* They dislike movies that are too sentimental.
* They dislike movies that are too light-hearted.

---

**SFT**

"Sherlock Holmes: A Game of Shadows" is an action-packed movie full of suspense and humor. If you're looking for a movie with strong performances from its leads, then this is the movie for you. It is a sequel to the 2009 film "Sherlock Holmes" and is set in the past. The movie is not too violent or dark, and it is not too long or slow-paced. It is the perfect movie for viewers who are looking for a light-hearted and entertaining film.

---

**P$^4$LM**

Sherlock Holmes: A Game of Shadows is a visually stunning action-packed thriller that will keep you on the edge of your seat. Robert Downey Jr. and Jude Law deliver strong performances as the legendary detective and his partner, Dr. Watson, as they track down the criminal mastermind Professor Moriarty. The film is full of suspense, humor, and thought-provoking themes, and it's sure to please fans of the original Sherlock Holmes stories.

---

In this instance, While the SFT's output is well-rounded and may appeal to those seeking light-hearted entertainment, the one generated by P$^4$LM is more likely to generate excitement and interest among a broader audience, including fans of the actors and the Sherlock Holmes series. Specifically, the latter provides specific details about the plot, the characters, and the actors' performances, which make it more engaging and informative. Also, mentioning Robert Downey Jr. and Jude Law leverages their star appeal, which is a significant draw for many viewers.

**Example 3:** (P$^4$LM achieving a much higher Preference Prioritization score than SFT)

---

**Movie: Scream (1996)**

Scream (1996) follows high school student Sidney Prescott, who becomes the target of a mysterious killer who wears a white mask and robes inspired by the Edvard Munch painting "The Scream". The killer uses horror films as part of a deadly game and becomes known as Ghostface. The film is notable for its self-referential deconstruction of the slasher film genre.

---

**User Preferences**

* They enjoy movies that are well-made and have strong performances.
* They appreciate movies that are visually appealing and have a strong sense of style.
* They like movies that are funny and have a light-hearted tone.
* They enjoy movies that are romantic and have a strong emotional core.
* They appreciate movies that are thought-provoking and have a strong social message.
* They dislike movies that are poorly made and have weak performances.
* They dislike movies that are visually unappealing and have a weak sense of style.
* They dislike movies that are unfunny and have a heavy-handed tone.
* They dislike movies that are unromantic and have a weak emotional core.
* They dislike movies that are unthought-provoking and have a weak social message.

---

**SFT**

Scream (1996) is a classic slasher film that is sure to keep you on the edge of your seat. The film follows Sidney Prescott, a high school student who becomes the target of a mysterious killer. The film is full of suspense and horror, and it is sure to leave you feeling unsettled. The film also features a strong female protagonist, which you are sure to appreciate.

---

**P⁴LM**

Scream (1996) is a self-referential slasher film that follows high school student Sidney Prescott, who becomes the target of a mysterious killer who wears a white mask and robes inspired by the Edvard Munch painting "The Scream". The film is notable for its dark humor, complex characters, and thought-provoking themes. It's also visually appealing and well-made, with a strong story and unique style. If you're looking for a horror film that's more than just a mindless slasher, Scream is a great choice.

---

Here, P⁴LM offers a more nuanced and appealing narrative for the given viewer profile. While the SFT output leans heavily into the thriller genre's conventional appeal with suspense and horror, it makes a somewhat narrow assumption about the viewer's preferences, focusing on the genre's intensity and a strong female lead without directly aligning with the viewer's interests specified in the profile. In contrast, P⁴LM asserts its appeal by highlighting the film's dark humor, thought-provoking themes, and a unique visual and storytelling style. These elements cater to the user's preferences for movies that are not only well-made and visually striking but also carry a strong emotional or intellectual impact. This endorsement suggests "Scream" transcends its genre's typical boundaries, offering depth and variety that align more closely with the user's select tastes, from humor and aesthetics to thematic substance, thereby making it the more compelling recommendation.

**Example 4:** (P⁴LM achieving a much higher Preference Coverage score than SFT)

**Movie: Ocean's Eleven (2001)**

Ocean's Eleven (2001) is a slick, stylish heist movie directed by Steven Soderbergh and starring George Clooney, Brad Pitt, Matt Damon, Don Cheadle, Andy Garcia, and Julia Roberts. The film, a remake of the 1960 Rat Pack film of the same name, follows a group of thieves and con artists who plan an elaborate robbery of three Las Vegas casinos. Ocean's Eleven, which oozes cool from its star-studded cast to its jazzy soundtrack, was a critical and commercial success, grossing $450 million worldwide.

**User Preferences**

* They enjoy movies that are light-hearted and fun.
* They appreciate movies with a strong sense of adventure.
* They like movies that are well-paced and exciting.
* They enjoy movies with a strong sense of humor.
* They appreciate movies with a unique and interesting story.
* They dislike movies that are dark and depressing.
* They dislike movies that are slow-paced and boring.
* They dislike movies that are too violent or gory.
* They dislike movies that are too sci-fi or fantasy-oriented.
* They dislike movies that are too sports-oriented.

**SFT**

Ocean's Eleven (2001) is a remake of the 1960 Rat Pack classic, following a group of thieves and con artists who plan an elaborate robbery of three Las Vegas casinos. With an all-star cast, a stylish director, and a jazzy soundtrack, it's a slick heist film that oozes cool. If you like dark, gritty, and violent movies, Ocean's Eleven may not be your first choice. However, if you're looking for a stylish, well-made heist film with a great cast, you'll probably enjoy it.

**P$^4$LM**

Ocean's Eleven (2001) is a slick, stylish heist movie that oozes cool from its star-studded cast to its jazzy soundtrack. The film follows a group of thieves and con artists who plan an elaborate robbery of three Las Vegas casinos. Ocean's Eleven is a well-made film with a strong story and plenty of action. It's also funny and entertaining, with a unique visual style. If you're looking for a movie that's visually appealing, well-made, and thought-provoking, Ocean's Eleven is a great choice.

The endorsement generated by P$^4$LM more effectively caters to the user's preferences by highlighting aspects of the movie that align closely with their interests. For example, it explicitly mentions the film being "funny and entertaining" and possessing a "unique visual style", which resonates with the viewer's appreciation for movies that are light-hearted, fun, and have a strong sense of humor. Additionally, by describing the movie as "well-made" with "plenty of action", it addresses the viewer's preference for films that are well-paced and exciting, as well as their interest in unique and interesting stories.

On the other hand, the output from SFT, while emphasizing the stylish and well-made nature of the film, might inadvertently deter the viewer by stating, "if you like dark, gritty, and violent movies, Ocean's Eleven may not be your first choice." This statement could be misconstrued as implying the film lacks in excitement, potentially conflicting with the user's aversion to dark and depressing movies. In contrast, P$^4$LM avoids any implication that the film might contain elements the user dislikes and affirms qualities they value, such as humor, visual appeal, and a captivating storyline.

## B Experimental Details

**SOTA Baselines** We compared the performance of our recommendation endorsement LMs with the following baselines:

1. **PaLM2-L**: We prompted PaLM2-L with item descriptions, user preference texts, and instructions to generate a response that suits the four endorsement principles.

2. **Supervised Fine-Tuned with Text (SFT-Text)**: We fine-tuned a PaLM2-XS with the aforementioned endorsement dataset but explicitly takes user-item texts as inputs.

3. **Supervised Fine-Tuned (SFT)**: We fine-tuned a PaLM2-XS model that utilizes user-item embedding vectors.

**Model-Based Evaluation Metrics** We evaluate the methods with a held-out unlabeled test dataset $\mathcal{D}_{\text{test}} = \{(\mathbf{I}^{(k)}, \mathbf{u}^{(k)})\}$, which consists of 96 (MovieLens) or 73 (Amazon) unique user and item pairs. Let $\theta_{RM} \in \{\text{Prec, App, Ppr, Pcov}\}$ denote a specific reward model used for scoring and $\Phi$ be the parameters of the LM. Then, we evaluate $\theta_{RM}(Y_{\Phi}^{(k)}; \mathbf{I}^{(k)}, \mathbf{u}^{(k)})$ for each sample in the test set and we report the average score per RM.

To better examine relative performances of the methods, we set SFT as the common baseline and compare the performance improvements of the other models against it. To this end, let $Y_{\text{SFT}}^{(k)}$ denote the response sampled by SFT given $(\mathbf{I}^{(k)}, \mathbf{u}^{(k)})$ as an input. Then, we compute the *win rate*, and *absolute increase* of an LM relative to $\{Y_{\text{SFT}}\}_{k=1}^{|\mathcal{D}_{\text{test}}|}$, which are defined as follows:

- **Win rate**:
  $\text{win\_rate}(\Phi; \theta_{RM}) =$

$$\frac{\sum_{k=1}^{|\mathcal{D}_{\text{test}}|} \mathbb{1}\left[\theta_{RM}(Y_{\Phi}^{(k)}; \cdot) > \theta_{RM}(Y_{\text{SFT}}^{(k)}; \cdot)\right]}{|\mathcal{D}_{\text{test}}|}$$

  where $Y_{\Phi}^{(k)}$ denotes the $k$th textual response sampled by the model $\Phi$.

- **Absolute increase**

$$= \frac{1}{|\mathcal{D}_{\text{test}}|} \sum_{k=1}^{|\mathcal{D}_{\text{test}}|} \left[\theta_{RM}(Y_{\Phi}^{(k)}; \mathbf{I}^{(k)}, \mathbf{u}^{(k)}) - \theta_{RM}(Y_{\text{SFT}}^{(k)}; \mathbf{I}^{(k)}, \mathbf{u}^{(k)})\right]$$

**Human Evaluation Metrics** Please check Appendix E for more information on the metrics and procedures for human evaluation.

### B.1 Details of Modeling and Pre-training

In this part, we discuss the details of modeling and pre-training process, focusing on both P⁴LM and SFT. We specifically elaborate on the integration of user and item behavioral embeddings into a unified latent space interpretable by a LM.

To facilitate the learning of this intricate mapping, we have conceptualized a series of tasks, orthogonal to the primary problem addressed in this study. First, note that to interpret embedding vectors, we require some semantic information about the entities to which they correspond. For instance:

- **Item embeddings:** Consider a movie(item) $i$ represented by its text-form description, denoted as $\mathbf{I}^{(i)}$. A supervised learning task is designed with the movie(item) embedding $\mathbf{i} \in \mathcal{V}$ as input and $\mathbf{I}^{(i)}$ as the target label. This approach enables the construction of varied tasks utilizing elements like critical reviews or movie(item) summaries to train the LM.

- **User embeddings:** A user $u$ is associated with a set of rated movies(items), $\mathcal{I}_u$. In other words, $\mathcal{I}_u = i : r_{u,i} \neq 0$. To textually describe a user, an LLM can be provided with the rating history $r_{u,i} \ \forall i \in \mathcal{I}_u$ to encapsulate the user's preferences. Given the extensive nature of $|\mathcal{I}_u|$, we selectively filter movies(items) and feed them to an LLM for summarization.[5]

  The user's rating history is then summarized into user preference profile text output $\{U_j(\mathbf{u})\}_{j=1}^J$ by the LLM. Consequently, a supervised learning task is developed with the user embedding $\mathbf{u} \in \mathcal{V}$ as the input and $\{U_j(\mathbf{u})\}_{j=1}^J$ as the corresponding target.

**Architecture** We construct our LM by augmenting a pre-trained transformer architecture T with additional *adapter layers* designed to map continuous behavioral embedding vectors to a common word embedding space. It's crucial to note that we are not training $\mathbf{u}$ or $\mathbf{i}$; rather, we focus on optimizing the adapter layers $W_I$ and $W_U$. This ensures that the nuanced information encapsulated in the CF embeddings in $\mathcal{V}$ is effectively translated into the word embedding space, $\mathcal{Z}$. Each of these adapter layers incorporates a 3-layer feed-forward network, interconnected with ReLU non-linearity. The conventional method is employed for mapping text tokens to word embedding space, whereas the adapter layers are utilized to map movie(item) and user embeddings to the latent space.

**Training Procedure** Our observations indicate that the simultaneous training of newly initialized adapter layers and the transformer parameters does not yield optimal results. This can be intuitively understood as the pretrained embedding layer has an established mapping to the language space, and the freshly initialized adapter layers necessitate extensive updates to achieve comparable mapping. To mitigate this challenge, we employ a two-stage training approach (Section 3). Initially, we exclusively train the adapters $W_U, W_I$ with the transformer parameters (T) set frozen, promoting more effective convergence in the subsequent stage. Following this, we proceed to fine-tune the complete model, engaging all the parameters of the LM. As an alternative, we can leverage parameter-efficient training approaches like the one proposed by Hu et al. (2021). This bifurcated training methodology proves pivotal in ensuring the convergence of LM.

## C    Data Generation

In this part, we discuss the data generation processes of various datasets used for evaluation. (1) We first elaborate on how we construct the dataset $\mathcal{D}$ used for supervised fine-tuning, as described in Section 3. Specifically, we describe the prompting techniques we used to generate the recommendation endorsement texts $Y$. (2) In order for an LM to interpret and utilize user and item behavioral information stored in their corresponding embedding vectors $(\mathbf{u}, \mathbf{i})$, we insert *adapter* layers $W_I$ and $W_U$.[6] These adapter layers map either $\mathbf{u}$ or $\mathbf{i}$ to the latent space that the LM can comprehend; that is, the token embedding space. We follow Tennenholtz et al. (2024) in training the adapter layers and the construction of the necessary datasets, which we detail below. (3) Finally, we also expand on how we have generated the AI labels for reward model training.

**Generation of the supervised fine-tuning dataset via self-critiquing and revision** Assuming we can represent a user/viewer with some text description, we can use various prompting techniques to query a very large LM to generate a personalized endorsement of an item to the user. To ensure that the endorsement outputted by the LM abides by the four

---

[5]In the MovieLens domain, we can safely assume that an LLM should have obtained vast implicit knowledge about most of the movies in the dataset, which is an unlikely assumption for the Amazon dataset. Hence, we first summarize the information of the items that a user has rated, and these descriptions are provided to an LLM along with the rating history.

[6]Note that the user-item behavioral embeddings $\mathbf{u}$, $\mathbf{i}$ can be obtained via a standard technique such as neural collaborative filtering Rendle et al. (2020) or a simpler matrix factorization Mnih & Salakhutdinov (2007).

principles set out in this work, we use *self-critiquing* and *self-revision* similar to the approach introduced in Bai et al. (2022).

In this approach, we first use few-shot prompting by querying PaLM2-L with an instruction, an item description $I^{(k)}$, and a user profile text $U_k(\mathbf{u}^{(k)})$ to generate a personalized endorsement text.

```
  Instructions:  Write a short and to-the-point personalized endorsement of a
movie in a single paragraph. You are given a synopsis of the movie and the user
preference profile information. Given these, the endorsement text should solely
based on the provided information.  That is, extra information not explicitly
mentioned in the provided synopsis should not appear in the endorsement. Assume
the viewer has not seen the movie yet. Hence, do not include any spoilers.  A
factual and personalized endorsement text is a brief statement that expresses
positive approval or recommendation of the film in a way that appeals to a specific
viewer, based on specific details or elements mentioned in its synopsis.
```

```
## Here follows few-shot examples. ##
```

```
...
```

```
Answer:
```

Then, the model generates the first endorsement text following the style presented in the few-shot examples. After that, we sample a *critique request* and query the model with the request, with the entire texts — including the just-generated endorsement — being given as the context. Here is the list of critique requests we used in the generation:

```
 Please comment on whether the last response in any ways appear to be not based
on the given synopsis, not aligned well with the viewer's preferences, or lacking
appeal. Keep your response in a single paragraph.
```

```
Does the last response seem off-base, not aligned with the viewer's preferences,
or lacking appeal? Please comment in a single paragraph.
```

```
Does the last response fail to be based on the synopsis, disregard the viewer's
preferences, or lack appeal? Please keep your response in a single paragraph.
```

```
Would you say that the last response is unrelated to the plot synopsis, misaligned
with the viewer's preferences, or lacking appeal?
```

```
Identify specific ways in which the last response is not factually consistent
with the provided synopsis. Keep your response in a single paragraph.
```

```
Discuss ways in which the endorsement deviates from the provided synopsis. Keep
your response in a single paragraph.
```

```
Imagine the only information about the movie is the one in the synopsis. Analyze
what additional information is discussed in the last response.
```

```
Identify if there are any hallucinated information in the last response compared
to the given synopsis.
```

```
Explain ways in which the last response may not align well with the preferences
of the viewer. Keep your response in a single paragraph.
```

```
Discuss whether the last response appeals to contradicting user preferences. Keep
your response in a single paragraph.
```

```
Identify in which ways the last response may be less appealing to the viewer.
Keep your response in a single paragraph.
```

```
Explain whether the last response tried too hard to meet all the viewer's
preferences. Keep your response in a single paragraph.
```

```
Discuss the last response and check whether it sounds sufficiently compelling as
an endorsement. Keep your response in a single paragraph.
```

```
Discuss whether the last response may contain spoilers of the movie. Keep your
response in a single paragraph.
```

```
Explain whether the last response would make someone to watch the movie. Keep
your response in a single paragraph.
```

```
Analyze the appeal and compellingness of the last response. Keep your response
in a single paragraph.
```

Our strategy of diversifying critique requests creates a dataset containing a wide range of responses in different styles. When a request is made, the model will autonomously assess its previous endorsement generation based on the specified criteria.

Following this evaluation, we instruct the model to revise its response in order to produce an endorsement that better aligns with the four criteria. Here are the prompts we used:

```
Please rewrite the response to remove any hallucinated information.
```

```
Please rewrite the endorsement such that all information is firmly grounded in
the synopsis.
```

```
Please revise the endorsement such that it is solely based on the information in
the synopsis.
```

```
Please rewrite the endorsement to refrain from mentioning information that cannot
be inferred from the synopsis.
```

```
Please rewrite the endorsement text to better fit the viewer's preferences.
```

```
Please rewrite the endorsement text such that it is coherent and consistent.
```

```
Please rewrite the endorsement text to look more personally appealing to the
viewer.
```

```
Please rewrite the endorsement text so that it appeals to specific parts of the
viewer's preferences.
```

```
Please rewrite the response to sound more appealing and compelling.
```

```
Please rewrite the response to remove any and all spoilers while keeping it sounds
appealing.
```

```
Please revise the response such that one would want to watch the movie after
reading the endorsement.
```

```
Revise the response to make it more appealing and compelling, while keeping it
in the same style.
```

```
Please rewrite the endorsement text to improve it for better precision (w.r.t.
the synopsis), better personalization, and better appeal.
```

```
Rephrase the endorsement text to make it more precise (in regards to the synopsis),
personalized, and appealing.
```

```
Please edit the endorsement text to make it more precise, personalized, and
appealing.
```

```
Please edit the endorsement text to make it more accurate, personalized, and
appealing.
```

**Data generation for the two-stage BC procedure as in Tennenholtz et al. (2024)** The adapter layers project user and item behavioral embeddings to the token embedding space of an LM, i.e., it is a mapping from $\mathcal{V}$ to $\mathcal{Z}$. In order to train these additional layers, we construct a dataset consisting of behavioral embeddings and their corresponding text descriptions. For example, to construct a user profile in text, a typical approach in conversational RS is to

provide the history of the user to an LLM as a prompt Kang et al. (2023); Salemi et al. (2024). For the MovieLens domain, we selected a few movies per each user based on the ratings. Then, we asked PaLM2-L to describe the characteristics of the user in a few bullet points. Here is an example:

---

**Example: user profile generation**

`Prompt:` In a few bullet points, describe the attributes and characteristics of a viewer who likesthe movies: Catch Me If You Can (2002), Cellular (2004), National Treasure (2004), DieHard 2 (1990), and The Matrix (1999) but dislikes the movies: Half Past Dead (2002), Predator (1987), In the Valley of Elah (2007), The Legend of Zorro (2005), and Mortal Kombat: Annihilation (1997).
`Sample Output:` * They enjoy movies that are fast-paced and action-packed. * They prefer movies with a strong plot and well-developed characters. * They appreciate movies with a sense of humor. * They are not interested in movies that are too violent or gory. * They do not enjoy movies that are too slow-paced or boring. * They prefer movies that are visually appealing. * They appreciate movies with a good soundtrack. * They are not interested in movies that are too predictable. * They enjoy movies that are thought-provoking and challenging.

---

Next, we employ the two-stage training procedure. In the initial stage, we fine-tune the adapter layers, and subsequently, we fine-tune the entire network. For additional details, we refer the readers to Tennenholtz et al. (2024).

**Generating AI feedback labels**    In Section 4.1, we explain how we establish and evaluate a score (i.e., reward) according to four key principles: Precision, Appeal, Preference Coverage and Preference Prioritization. For the Appeal, Preference Coverage, and Preference Prioritization, we employ the LAIF methodology as suggested in Lee et al. (2023). Although, in principle, it is feasible to use AI feedback for training the Prec RM, we choose to use an existing off-the-shelf NLI model that has been trained on a substantially larger dataset of textual entailment examples. The other RMs, however, are tailored to our specific application. Therefore, we use another LM to generate and train these RMs in accordance with the loss functions outlined in the main text.

Below is the example prompt for eliciting AI feedback on Preference Prioritization:

```
[QUESTION]
Options:
(A). [ENDORSEMENT A]
(B). [ENDORSEMENT B]
User's preferences: [USER_PROFILE]
Let's think step-by-step: [CHAIN_OF_THOUGHT]
Answer: [ANSWER]
```

In our approach, we select a question from a predetermined question pool and repeat this process multiple times for each example. This method is designed to obtain more robust labels from the model. The labels for the two responses are derived directly from the supervised fine-tuning dataset, as previously described. Additionally, in line with the findings of Bai et al. (2022), we observed that using hard binary labels tends to lead to overfitting in the developed RMs. To mitigate this, we utilize the average normalized probabilities of the two responses as labels Kadavath et al. (2022). This average is calculated across the ensemble of sampled questions.

Below, we present the lists of questions we ask for Appeal, Preference Coverage and Preference Prioritization RMs, respectively. We show the examples for the MovieLens domain, but the adaptation to the Amazon product review domain is straightforward.

---

**Questions for eliciting Preference Prioritization labels from PaLM2-L**

1. Which of the following two responses resonates more deeply with the key elements of the viewer's preferences?

2. Choose the response that engages more effectively with specific aspects of the viewer's preferences.

3. Select the option that appears to delve more deeply and connect more meaningfully with certain aspects of the viewer's interests.

4. From the two options presented, which one do you think dives deeper into aligning with the viewer's specific preferences?

5. Decide which of these two alternatives more effectively hones in on and matches key aspects of the viewer's preferences.

6. Identify the option out of these two that best delves into and reflects the core interests of the viewer.

7. Pick the response that appears to engage more thoughtfully with the viewer's specific tastes.

8. Among the given choices, select the one you feel offers a more insightful connection with what the viewer strongly prefers.

9. Which option, out of the two provided, seems to engage more deeply with the viewer's personal preferences?

10. Choose the response that you think demonstrates a more focused alignment with the viewer's individual preferences.

11. Determine which of these two choices appears to be more engaging and relevant to the specific likes and interests of the viewer.

12. From the two given options, identify the one that seems to offer a more nuanced and focused approach tailored to meet the viewer's unique preferences.

---

**Questions for eliciting Appeal labels from PaLM2-L**

1. Which of the following two responses has more appeal and reads more compelling?

2. Choose one of the following two options that sounds more compelling and persuasive.

3. Please select the option that you find more engaging and convincing from the two provided.

4. Between these two options, which one strikes you as more convincing and engaging?

5. Can you determine which of the two given responses holds greater appeal and is more compelling?

6. Of the following two possibilities, which one do you find more persuasive and appealing?

7. From the two options presented, which one do you believe is more compelling and persuasive?

8. Considering the two options below, which one appeals to you more as being persuasive and compelling?

9. Please indicate which of these two options you perceive as more engaging and convincing.

10. Please evaluate and choose the more compelling and persuasive option from the two presented.

11. Reflect on the two provided responses and select the one that you find more engaging and persuasive.

12. Analyze these two options and decide which one you believe to be more compelling and persuasive. preferences.

**Questions for eliciting Preference Coverage labels from PaLM2-L**

1. Between these two responses, which one more comprehensively addresses the diverse preferences of the viewer?

2. Which of these two responses provides broader coverage of the viewer's preferences?

3. Considering the viewer's profile, which of the two movie endorsements more comprehensively addresses their listed preferences?

4. Given the specific preferences in the viewer's profile, which endorsement text among A and B aligns more closely with the majority of these preferences?

5. In relation to the viewer's detailed profile, which of the endorsement texts demonstrates greater coverage of the viewer's movie preferences?

6. Taking into account the viewer's preferences, which endorsement more effectively caters to a broader range of their interests in movies? Select between A and B.

## D   Fine-tuning LMs with Reinforcement Learning

Recall the LM $\Phi\big(Y = \{y_n\}_{n=0}^{N-1} \mid y_0; \mathbf{I}, \mathbf{i}, \mathbf{u}\big) = \prod_{n=0}^{N-1} \Phi\big(y_n \mid y_{0:n-1}; \mathbf{I}, \mathbf{i}, \mathbf{u}\big)$ with item text $\mathbf{I}$, item and user CF embedding vectors $(\mathbf{i}, \mathbf{u})$ and the reward model $r(Y, \mathbf{I}, \mathbf{i}, \mathbf{u})$ that measures the quality of factuality, appeal, preference prioritization, and preference coverage of a given

recommendation endorsement. Also recall the generation process of LMs can be modeled using the following $N$-horizon context MDP:

$$c = (\mathbf{I}, \mathbf{i}, \mathbf{u}), \quad s_n = y_{0:n-1}, \quad a_n = y_n, \quad s_0 = y_0, \quad P(s_{n+1} \mid s_n, a_n) = \delta\{s_{n+1} = (s_n, a_n)\},$$

$$r(s_n, a_n; c) = \begin{cases} r(s_{n+1}; c) = r(y_{0:n}; \mathbf{I}, \mathbf{i}, \mathbf{u}) & \text{if } n = N-1 \\ 0 & \text{otherwise} \end{cases}, \quad \pi(a_n \mid s_n; c) = \Phi(y_n \mid y_{0:n-1}; \mathbf{I}, \mathbf{i}, \mathbf{u}),$$

where $\delta_z$ denotes the Dirac distribution at $z$. As a result, optimizing RL policy $\pi$ is equivalent to fine-tuning the underlying LM. The system starts from the start-of-sentence token $y_0$, equipped with user-item context $c$. Given the MDP state $s_n$, the policy takes the action at token-step $n$ as the next generated token $y_n$. As a result of this action, the system transition deterministically to the state which corresponds to the updated token sequence. The reward is zero, except at the final step in which measures the overall quality of the texts at the end of the auto-regressive generation process.

A common goal in fine-tuning the LM is to maximize the average over-all quality of the generated text response given the context distribution, *i.e.*, $\max_\Phi \quad \mathbb{E}_{(\mathbf{I},\mathbf{i},\mathbf{u})} \mathbb{E}_{\Phi(y_{0:N-1}|\mathbf{I},\mathbf{i},\mathbf{u})}[r(Y; \mathbf{I}, \mathbf{i}, \mathbf{u})]$. The gradient of this objective function can be obtained as follows: $\nabla_\Phi \mathbb{E}_{(\mathbf{I},\mathbf{i},\mathbf{u})} \mathbb{E}_{\Phi(y_{0:N-1}|\mathbf{I},\mathbf{i},\mathbf{u})}[r(Y; \mathbf{I}, \mathbf{i}, \mathbf{u})] = \mathbb{E}_c \mathbb{E}_{\pi(\cdot|s_{0:N};c)} \left[ r(s_N; c) \sum_{n=0}^{N-1} \nabla_\Phi \log \pi(s_n|a_n; c) \right]$. This is equivalent to applying the popular policy gradient algorithm REINFORCE to the aforementioned context MDP for endorsement text generation. The gradient of the objective function is estimated using trajectories $\prod_{n=0}^{N-1} \pi(s_n|a_n; c)$ generated by the current policy, and then used to update the LM policy in an online fashion.

**Adding KL regularization:** The risk of fine-tuning purely based on the reward model learned from human or AI feedback is that it may overfit to the reward model and degrade the "skill" of the initial LM. To avoid this phenomenon, similar to (Ouyang et al., 2022; Stiennon et al., 2020), we add the KL between the fine-tuned and pre-trained models as a regularizer to the objective function. Leveraging the auto-regressive nature of LMs one can compute the KL regularization over the entire sequence/trajectory (of tokens), i.e., $\mathrm{KL}\big(\Phi(y_{0:N-1}|\mathbf{I}, \mathbf{i}, \mathbf{u}) \| \Phi_{\mathrm{BC}}(y_{0:N-1}|\mathbf{I}, \mathbf{i}, \mathbf{u})\big)$. The resulting objective function is as follows:

$$\max_\Phi \; J(\Phi) := \mathbb{E}_{(\mathbf{I},\mathbf{i},\mathbf{u})} \mathbb{E}_{\Phi(y_{0:N-1}|\mathbf{I},\mathbf{i},\mathbf{u})} \left[ r(y_{0:N-1}; \mathbf{I}, \mathbf{i}, \mathbf{u}) - \beta \log \frac{\Phi(y_{0:N-1}|\mathbf{I}, \mathbf{i}, \mathbf{u})}{\Phi_{\mathrm{BC}}(y_{0:N-1}|\mathbf{I}, \mathbf{i}, \mathbf{u})} \right]. \quad (3)$$

It can be shown that this problem is equivalent to the KL-regularized objective in the CoMDP.

Denote by $\mathcal{D}$ a replay buffer of trajectories $\{(\mathbf{I}, \mathbf{i}, \mathbf{u}, y_{0:N-1})\}$ generated by arbitrary "off-policy" LMs $\Phi'(y_{0:N-1}|\mathbf{I}, \mathbf{i}, \mathbf{u})$ (e.g., the LM $\Phi'$ does not necessarily equal to the "on-policy" LM $\Phi$) over various contexts $(\mathbf{I}, \mathbf{i}, \mathbf{u})$. Below we aim to leverage the abundance of offline text-token sequence trajectories for more efficient LM policy learning. Denote by $\tau = \{(c, s_n, a_n, s_{n+1})\}_{n=0}^{N-1} \sim \mathcal{D}$ a trajectory sampled from the offline data $\mathcal{D}$, where $(s_n, a_n, s_{n+1})$ is a tuple of state, action, and next state of the context MDP, respectively. The addition of KL regularization (Haarnoja et al., 2018; Carta et al., 2021), which was originally intended to avoid overfitting to the reward model and discounting the "skill" of the initial LM, has also been shown to alleviate the out-of-distribution action data generalization issues arisen from off-line RL (Kumar et al., 2019). With this KL regularization we can utilize the *soft actor critic* framework (Haarnoja et al., 2018) to develop RL updates for the *value function* $\{V_n(s; c)\}_{n=0}^{N-1}$, *state-action value function* $\{Q_n(s, a; c)\}_{n=0}^{N-1}$, and *LM policy* $\prod_{n=0}^{N-1} \pi(s_n|a_n; c)$ (initialized with

$\prod_{n=0}^{N-1} \pi_{BC}(s_n|a_n;c))$ that minimizes the following losses:

$$L_Q = \mathbb{E}_{\tau \sim \mathcal{D}} \left[ \sum_{n=0}^{N-2} (V_{tar,n+1}(s_{t+1};c) - Q_n(s_n,a_n;c))^2 + (r(s_N;c) - Q_{N-1}(s_{N-1},a_{N-1};c))^2 \right], \quad (4)$$

$$L_V = \mathbb{E}_{\tau \sim \mathcal{D}} \left[ \sum_{n=0}^{N-1} (Q_{tar,n}(s_n,a_n;c) - \alpha \log \frac{\pi(a_n|s_n;c)}{\pi_{BC}(a_n|s_n;c)} - V_n(s_n;c))^2 \right], \quad (5)$$

$$L_\pi = \mathbb{E}_{\tau \sim \mathcal{D}} \left[ \sum_{n=0}^{N-1} Q_n(s_n,a_n;c) - \alpha \log \frac{\pi(a_n|s_n;c)}{\pi_{BC}(a_n|s_n;c)} \right], \quad (6)$$

where the critic $Q_n$ and $V_n$ take any token sequences at step $n$ as input and predict the corresponding cumulative return; $\alpha > 0$ is the entropy temperature; $(V_{tar,n}, Q_{tar,n})$ are the target value networks.

Besides iteratively updating the LM policies and their critic functions, consider the closed-form optimal solution of the Bellman equation of this entropy-regularized RL problem:

$$V_n^*(s;c) = \alpha \cdot \log \mathbb{E}_{a \sim \pi_{BC}(\cdot|s;c)} [\exp(\frac{Q_n^*(s,a;c)}{\alpha})], \forall n, \quad (7)$$

$$Q_{N-1}^*(s,a;c) = r(s;c), \; Q_n^*(s,a;c) = \mathbb{E}_{s' \sim P(\cdot|s,a)} [V_{n+1}^*(s';c)], \forall n < N-1, \quad (8)$$

$$\mu_n^*(a|s;c) = \pi_{BC}(a|s;c) \cdot \exp(\frac{Q_n^*(s,a;c)}{\alpha}) / \mathbb{E}_{a \sim \pi_{BC}(\cdot|s;c)} [\exp(\frac{Q_n^*(s,a;c)}{\alpha})], \forall n, \quad (9)$$

where the time-dependent optimal policy (at time $n$), i.e., $\mu_n^*$ is a softmax policy w.r.t. the optimal state-action values $Q_n^*$ over different actions sampled from the pre-trained LM $\pi_{BC}$. Therefore, a value-based approach for RL-based LM fine-tuning would be to first learn the optimal value functions $\{Q_n^*\}$ via the Bellman residual minimization procedure (Antos et al., 2008) applied to Eq. (7) and Eq. (8) and then solve the following policy distillation (Czarnecki et al., 2019) problem: $\pi \in \arg\min_\pi \mathbb{E}_{\tau \sim \mathcal{D}} \left[ \sum_{n=0}^{N-1} \mathrm{KL}(\pi(\cdot|s_n;c) || \mu_n^*(\cdot|s_n;c)) \right]$ with respect to the optimal value $\{Q_n^*\}$. Notice that this amounts to updating the LM model $\pi$ via the gradient update

$$\Phi \leftarrow \Phi - \gamma \cdot \mathbb{E}_{\tau \sim \mathcal{D}} \left[ \sum_{n=0}^{N-1} \mathbb{E}_{a \sim \pi(\cdot|s;c)} \left[ \nabla_\Phi \log \pi(a|s;c)(\log \frac{\pi(a|s;c)}{\pi_{BC}(a|s;c)} - \frac{Q_n^*(s,a;c)}{\alpha}) \right] \right], \quad (10)$$

with learning rate $\gamma > 0$. Further techniques in value-function parameterization have been employed to tackle the overestimation bias. (Fujimoto et al., 2018) proposed maintaining two $Q$ functions, and a *dual Q* function chooses the minimum value between them to avoid overestimation. (Jaques et al., 2019) applies dropout in the $Q$ function to maintain an *ensemble* of $Q$ values, and outputs the minimum value to avoid overestimation.

## E  Rater Evaluation

Our test set comprised 96 instances from the MovieLens dataset and 73 instances from the Amazon dataset. For the MovieLens dataset, evaluators were presented with a movie synopsis alongside two endorsements for comparison.[7] Their task was to determine which endorsement was more accurate and which was more appealing, as illustrated in Figure 8. For the Amazon dataset, evaluators were shown a description of an item accompanied by two endorsements. In a manner akin to the win rate metric introduced in Appendix B for MB evaluation, we calculate the ratio at which model outputs were preferred over the baseline outputs (i.e., SFT). More concretely,

$$\text{win\_rate}(\Phi_i \succ \Phi_{SFT}) = \frac{\sum_{k=1}^{|\mathcal{D}_{test}|} \mathbb{1} [Y_{\Phi_i}^{(k)} \succ Y_{SFT}^{(k)}]}{|\mathcal{D}_{test}|}, \quad (11)$$

---

[7]To minimize bias in the evaluation by raters, we randomized the presentation order of the two endorsements.

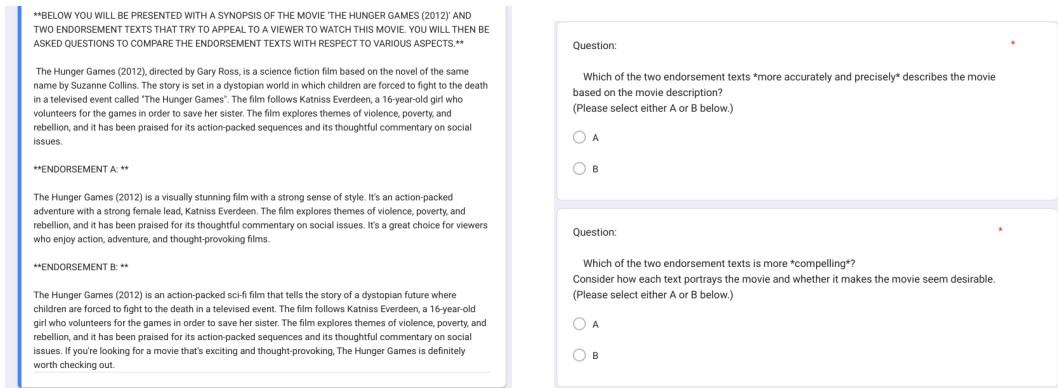

Figure 8: Sample Form for Human Rater Evaluation for Pairwise Comparison

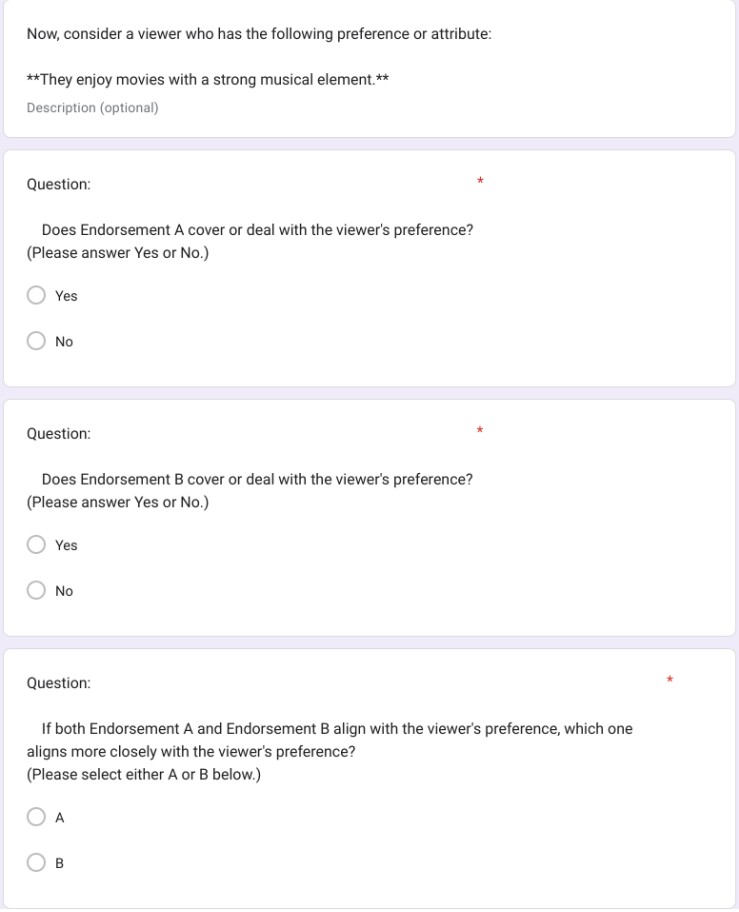

Figure 9: Sample Form for Human Rater Evaluation for Preference Coverage and Prioritization

where $Y^{(k)}_{\Phi_i}$ denotes the $k$th item endorsement sampled by the model $\Phi_i \in \{\text{PaLM2-L, SFT-Text, P}^4\text{LM}\}$, and $Y^{(k)}_{\text{SFT}}$ is the corresponding endorsement generated by SFT.

To assess Preference Prioritization and Preference Coverage, we developed specific questions aimed at more accurately quantifying relative preferences. Remember that the user profile

text we use consists of a bullet-point list detailing various user attributes and preferences. Initially, in our human evaluation process, we presented raters with the complete user profile text, asking them to determine which endorsement better aligned with the user's preferences. However, this approach yielded noisy and unreliable responses, and feedback indicated that the raters found it challenging to process the extensive information provided.

To address this issue, we refined our approach by selecting a subset of user attributes from the bullet-point list in each user profile. We then asked raters to evaluate these attributes individually. Specifically, we inquired if each of the endorsement texts addressed the presented user attribute and, if so, which one did so more effectively (see Figure 9). For Preference Prioritization, we asked raters to directly compare the outputs from our model with baseline outputs, and we calculated the proportion of instances where our model's outputs were preferred (i.e., win rate). For Preference Coverage, we compared how often the chosen user attributes were addressed in both the model and baseline outputs. We then quantified the improvement of our model's outputs over the baseline as a percentage.

Formally, let $U(\mathbf{u}^{(k)}) = \{U_j(\mathbf{u}^{(k)})\}_{j=1}^{|J|}$ represent the user preference texts for the user in the $k$th example of the test set, which includes $|J|$ bullet points with $J$ being the corresponding index set. We randomly select a subset of $|M|$ attributes from this list, forming an index set $M^{(k)}$. For each attribute $m \in M^{(k)}$, we inquire raters if the attribute is addressed by both $\Phi_i$ and SFT. Subsequently, we tally the instances where test model outputs and SFT outputs encompass the chosen attributes throughout all examples in the test set. Following this, the Pcov score is calculated as the percentage increase in the frequency of attributes covered by the model compared to those covered by the SFT. That is,

$$\text{Pcov}(\Phi_i) = \frac{\sum_{k=1}^{K} \sum_{m \in M^{(k)}} \mathbb{1}\left[Y_{\Phi_i}^{(k)}(A_m) = 1\right] - \sum_{k=1}^{K} \sum_{m \in M^{(k)}} \mathbb{1}\left[Y_{\Phi_{\text{SFT}}}^{(k)}(A_m) = 1\right]}{\sum_{k=1}^{K} \sum_{m \in M^{(k)}} \mathbb{1}\left[Y_{\Phi_{\text{SFT}}}^{(k)}(A_m) = 1\right]}, \quad (12)$$

where $\mathbb{1}\left[Y_{\Phi}^{(k)}(A_m) = 1\right]$ denotes that the $k$th model output addresses the user attribute $A_m$ sampled from $\{U_j(\mathbf{u}^{(k)})\}_{j \in M}$.

As illustrated in Figure 9, raters were also tasked with identifying which of the two endorsement texts more closely matched the given user attribute. For calculating the Preference Prioritization score, we only considered instances where raters indicated that both endorsements addressed an attribute. The preferences expressed by raters were then used to determine the win rate of a test model compared to SFT, employing the same metric outlined in equation 11.

## F  Prompts

We put the prompts that we used when evaluating the RMs as discussed in Appendix A.1.

---

**Prompt to Gemini Ultra for Evaluating the Precision (Prec) RM**

Craft two endorsements of the movie [TITLE] whose plot is provided below:
[PLOT]
he endorsement should be factually consistent, relying solely on the information from the plot. Avoid copying the plot verbatim. The endorsement should be a single paragraph. Do NOT use any information that is not explicit in the plot, even if you know other fact about the movie.
Endorsement: [GOOD_ENDORSEMENT]
Now create another endorsement for the same movie. This new endorsement should be less factually consistent with the plot. That is, this endorsement should have a lower quality than the previous in terms of precision. Also, feel free to hallucinate, or to add some untrue fact in the endorsement w.r.t. the plot. It should be easy to tell this endorsement is worse than the previous. The length of this endorsement should be similar to the length of the previous one.
Non-Precise Endorsement: [BAD_ENDORSEMENT]

---

**Prompt to Gemini Ultra for Evaluating the Preference Prioritization (Ppr) RM**

Craft a personalized endorsement of the movie [TITLE] whose plot is provided below:
[PLOT]
The endorsement should be personalized to a user with the following preferences:
[USER_PROFILE]
The endorsement should be factually consistent, relying solely on the information from the plot. Avoid copying the plot verbatim. The endorsement should be a single paragraph. Do NOT use any information that is not explicit in the plot, even if you know other fact about the movie.
Tailor the endorsement to align with the user's preference profile, highlighting aspects of the movie that would particularly appeal to them. Ensure each appeal is directly tied to the plot elements.
Endorsement: [GOOD_ENDORSEMENT]
Now create another endorsement for the same movie. This endorsement should be less personalized to the user's profile. It should be easy to tell this endorsement is worse than the previous in terms of personalization.
Less-Personalized Endorsement: [BAD_ENDORSEMENT]

---

**Prompt to Gemini Ultra for Evaluating the Appeal (App) RM**

Craft an appealing and compelling endorsement of the movie [TITLE] whose plot is provided below:
[PLOT]
The endorsement should be factually consistent, relying solely on the information from the plot. Avoid copying the plot verbatim. The endorsement should be a single paragraph. Do NOT use any information that is not explicit in the plot, even if you know other fact about the movie.
Ensure each appeal is directly tied to the plot elements. Make the endorsement as appealing and compelling as possible.
Appealing Endorsement: [GOOD_ENDORSEMENT]
Now create another endorsement for the same movie. This endorsement should be less appealing and compelling than the first one. It should be easy to tell this endorsement is worse than the previous in terms of appealingness and compellingness.
Less-Appealing Endorsement: [BAD_ENDORSEMENT]

---

**Prompt to Gemini Ultra for Evaluating the Preference Coverage (Pcov) RM**

Craft a personalized endorsement of the movie [TITLE] whose plot is provided below:

[PLOT]

The endorsement should be personalized to a user with the following preferences:

[USER_PROFILE]

The endorsement should be factually consistent, relying solely on the information from the plot. Avoid copying the plot verbatim. The endorsement should be a single paragraph. Do NOT use any information that is not explicit in the plot, even if you know other fact about the movie.

Tailor the endorsement to align with the user's preference profile, highlighting aspects of the movie that would particularly appeal to them. Ensure each appeal is directly tied to the plot elements.

Importantly, make sure the endorsement covers both positive and negative preferences w.r.t. the plot. For example, if the user does not like action movies, then the endorsement should cover the fact that the movie is an action movie.

Preference Covering Endorsement: [GOOD_ENDORSEMENT]

Now create another endorsement for the same movie. This endorsement should not cover all of the user's preferences like the first one. It should be easy to tell this endorsement is worse than the previous in terms of preference coverage..

Less-Covering Endorsement: [BAD_ENDORSEMENT]

