# OpenReview forum: "Factual and Tailored Recommendation Endorsements using Language Models and Reinforcement Learning"
_colmweb.org/COLM/2024/Conference — COLM_

### Official Review · Reviewer_kDyJ · 2024-04-22

**Rating:** 8
**Confidence:** 5
**Ethics Flag:** 1

**Summary:**

In this paper, the authors develop a novel P4LM model to endorses recommended items by emphasizing specific item characteristics and their coverage to a user’s preferences. It takes user & item embeddings as inputs to generate responses that are appealing, factually-grounded and tailored to the user’s preferences. P4LM employs a joint reward function as AI-based feedback for reinforcement learning. Through extensive experiments on offline datasets, the authors demonstrate that P4LM delivers more appealing and tailored endorsements to users, as determined by auto-critic and rater evaluations.

Overall, the paper is very nicely written, the research idea is novel and technically sound, and I especially commend the authors for conducting a comprehensive set of ablation studies to demonstrate the generalizability of their proposed method. I will definitely recommend acceptance of this paper. Meanwhile, I also offer a few suggestions below, which I hope the authors will find them useful:

(1)	I wonder what is the exact role of LM in the designed method Since the authors directly inject the embeddings into LM (a seq2seq LM in particular) and generates the response directly, it seems that LM is equivalent to a hidden interaction layer between user & item in this case.

(2)	I also wonder the improvements of the endorsement quality (demonstrated through a couple of evaluation metrics in this paper) will or will not eventually lead to the improvements of business metrics related to recommendations (CTR for example).

**Questions To Authors:**

Please refer to the summary section.

**Reasons To Accept:**

Please refer to the summary section.

**Reasons To Reject:**

Please refer to the summary section.

---

> ### Author Rebuttal · Authors · 2024-05-31
>
> Thank you for recognizing the value of our work and for your constructive feedback. We appreciate your insightful comments and suggestions.
>
> 1. __what is the exact role of LM in the designed method__
>
> Thanks for the interesting comment. Although we directly inject the embeddings into the LM, without using a pretrained LLM, we cannot guarantee the language quality of responses generated by the trained model. The generalization capability of pretrained LLMs helps improve response quality even for specific downstream domains, as observed in prior work.
>
> In domains with large, diverse, high-quality domain-specific textual datasets, training an LLM from scratch followed by RLAIF training may be beneficial. However, using a pretrained (or instruction-tuned) LLM will likely significantly enhance language quality. The tradeoff may include potential hallucinations, as information from the pre-training dataset can be ingrained in the model, causing it to include information not supported in the given context in responses. In this case, P4LM's training framework, which ensures factually grounded responses through the Precision RM, can mitigate the impact of hallucinations.
>
> 2. __the improvements of the endorsement quality will or will not eventually lead to the improvements of business metrics__
>
> This is indeed a very interesting question. We plan to explore this as future work, where P4LM can be integrated into a deployed ConvRec system to generate endorsement text for the items selected by the recommendation engine. For the scope of this work, we focus on the endorsement task itself, isolating this problem to better evaluate endorsement quality through extensive model-based and human evaluations.

---

### Official Review · Reviewer_wcse · 2024-05-03

**Rating:** 8
**Confidence:** 5
**Ethics Flag:** 1

**Summary:**

# Summary

The paper focuses on generating good natural language endorsements for recommendations provided to a user. They introduce four criteria: 1) factuality (referred to as precision) 2) appeal 3) coverage of user's preferences and 4) prioritization of aspects that matter most to the user. They quantify these criteria, learn a reward model for each, and then use RLAIF to align LM with the criteria.

A separate contribution of the work is to learn projections from CF embeddings of users and items to a common space with the token embeddings, so the CF embeddings can be used directly in the model instead of having to generate natural language profiles of users.

They compare their model with 1) a bigger LLM prompted with text only to generate a response adhering to the four criteria 2) a text-only smaller SFT model and 3) a smaller SFT model fine-tuned with embeddings. Their results show that their RLAIF trained model does better than the other models evaluated.

# Overall evaluation

* The paper is written clearly and is easy to follow. The data collection and experimental evaluation makes sense.
* Decomposition of the objective into the four criteria makes intuitive sense, but I would have liked to see more evaluation, more details below.
* The part about injecting CF embeddings directly into the model seems like a tangent from the main point of the paper, more below.
* I do have an issue with the use of the word precision for the criteria of factuality. Precision has a very specific meaning as a metric in the IR community, it'll be good to use a different word here.

**Questions To Authors:**

* Consider using a different term for factuality than precision.

**Reasons To Accept:**

* The paper is clearly written and is good technically. The choices made for the different reward models make sense.
* Decomposition of the reward into easily understandable criteria makes it easier to train and debug when a model is producing unsatisfactory results.
* Human evaluation is performed in addition to the model based evaluation, which increases my trust in the claims.
* The experimental plan, though restricted to a single family of LLMs, is mostly appropriate for the task.

**Reasons To Reject:**

* The whole section about user profile prompting is tangential to the main thrust of the paper. If it is included, it needs to be evaluated as well. E.g. an RLAIF trained PALM2-XS model that uses text based user representation instead of embedding.
* There is some conflict between criteria 3 and 4, e.g. if we prioritize focus on some aspects (criteria 4), this might mean leaving some aspects out, which would reduce coverage of user's preferences (criteria 3). Could the authors' talk a bit more about this?
* In the experimental section, the human evaluation and reward based evaluation have some discrepancies (attributed to approximation errors in the reward models). This needs to be discussed more, as noise in the reward models impacts the whole RLAIF methodology.
* Finally, it's not clear to me if the division into the four criteria actually helps increases the overall quality of the endorsements. It would've been good to have human evaluation for the overall endorsements and establish that we see improvement in this score as well.

---

> ### Author Rebuttal · Authors · 2024-05-31
>
> Thank you for your thoughtful and constructive feedback. We appreciate your insights and have addressed each point as follows:
>
> 1. __CF embeddings...__
>
> Our main contribution is identifying the four criteria for recommendation endorsements and using RLAIF to achieve them. Defining optimal textual user profiles is beyond the scope of this work. In this regard, mapping from a CF embedding space to a word-token embedding space allows us to use diverse sources of user data, reducing reliance on specific prompts. The output from adapter $W_{\mathcal{U}}$ compactly represents collaborative-filtering-based and other user details, minimizing dependence on specific profile formatting.
>
> 2. __text-based RLAIF baseline__
>
> We agree that text-based RLAIF is an important baseline. However, we aimed to avoid relying on specific user profile formats. Preliminary trials showed this baseline struggled to maximize the rewards, possibly due to increased context lengths. We are re-evaluating this baseline and will include the results in the final version.
>
> 3. __the word precision__
>
> We understand the concern, and we will adopt a different term in our revised paper.
>
> 4. __conflict between criteria 3 and 4__
>
> Although we acknowledge a potential conflict between these criteria, these scores are positively correlated in practice. Tables 6 and 7 show that training with only the Ppr or Pcov RM resulted in high scores for both. This is because Pcov RM training does not exclude prioritization. We will expand on this in the final version.
>
> 5. __human vs. model-based (MB) evaluation__
>
> As noted on page 8, MB and human evaluations align well on Amazon, but differ on MovieLens. Raters preferred P4LM much more w.r.t. Pcov, Ppr, and App scores than MB evaluations suggest. This likely arises because these RMs were trained with PaLM2-L data. Using a different LLM for data generation might help, but won't fully resolve the issue. We will evaluate RM alignment with raters' responses, comparing RM and rater preferences for each endorsement text.
>
> 6. __overall improvement by P4LM__
>
> We do have overall quality comparison from the raters (ratios of preferred responses over SFT's):
> | Amazon | Ratio |
> |-|-|
> | PaLM2-L     | 0.466 |
> | SFT-Text    | 0.438 |
> | Ours | 0.808 |
>
> | MovieLens | Ratio |
> |-|-|
> | PaLM2-L | 0.333  |
> | SFT-Text | 0.490 |
> | Ours | 0.708 |
>
> These results show significant overall quality improvement with P4LM as evaluated by human raters. We will include these in the final version.

---

> > ### Comment · Reviewer_wcse · 2024-06-03
> >
> > I thank the authors for the detailed response. I have updated my scores assuming the final version will be updated with all these results. Please also include the challenges you mentioned around maximizing rewards with text-based user profiles, even without evaluation numbers, this will be useful information for others working in this area, thanks!

---

### Official Review · Reviewer_VTxA · 2024-05-08

**Rating:** 6
**Confidence:** 4
**Ethics Flag:** 1

**Summary:**

The paper introduces innovative reward models designed to quantify key elements crucial for effective personalization. By employing Reinforcement Learning with Augmented Intrinsic Rewards (RLAIF) and these novel reward structures, The Language Model (LM) generates item endorsements that are not only factual and appealing but also comprehensive in addressing user preferences. Furthermore, the prioritization mechanism enhances the connection between relevant item attributes and user preferences, boosting the likelihood of users accepting high-value recommendations.

**Questions To Authors:**

See the weakness above.

**Reasons To Accept:**

1. Use the collaborative information of users and products to join the large model and fine-tune it, and use a two-stage training method to enable better integration.
2. Four aspects of criteria are used to measure the quality of the results, and an RLAIF technique is designed to fine-tune the model.
3. The model and formula derivation part is quite clear, and the specific details are included in the appendix.

**Reasons To Reject:**

1. In the experimental part, apart from SFT fine-tuning, only compare PaLM2. There are already many articles on combining large models with user information. I think it is very important to compare with them. In addition, the model should be compared with one or two traditional models.
2. The abscissa in Figure 2 is not shown and the ordinate on the right side is not shown. The four indicators should be put into a common coordinate axis in one picture to display, which will make it clearer.
3. The main experiments of the article should be reflected in the main text, such as ablation experiments, etc., and should not be found in the appendix. It gives people a feeling that the things in the main text are not as important as in the appendices.
4. Some figures and tables are not mentioned at all in the text, such as Figure 2. Moreover, there are too many references in the appendix of the article, and no anonymous code link is published.

---

> ### Author Rebuttal · Authors · 2024-05-31
>
> Thanks for your valuable feedback. Please find brief responses below.
>
> 1. __many articles on combining large models with user information … should be compared with one or two traditional models.__
>
> Indeed, there is recent work that combines user information with large language models (e.g., [1, 2, 3], listed below). __However, these papers focus on the recommendation task itself, whereas our work addresses the novel task of generating endorsement text for recommended items that satisfy the criteria of precision, appeal, preference coverage, and prioritization.__ As such, these methods do not admit a direct comparison with ours. Traditional recommenders are also unsuitable for our task as they do not generate natural language responses.
>
> [1] CoLLM: Integrating Collaborative Embeddings into Large Language Models for Recommendation (Zhang et al., 2023)
>
> [2] Representation Learning with Large Language Models for Recommendation (Ren et al., 2024)
>
> [3] Collaborative Large Language Model for Recommender Systems (Zhu et al., 2024)
>
> 2. __The abscissa in Figure 2 is not shown...__
>
> We appreciate the feedback on Figure 2 and will revise it to include a clear abscissa and ordinate, as well as combine the four indicators on a common coordinate axis to enhance clarity.
>
> 3. __The main experiments of the article should be reflected in the main text, such as ablation experiments, etc., and should not be found in the appendix…__
>
> We acknowledge the importance of having main experiments in the primary text. Unfortunately, due to the strict 9-page limit excluding references, we were forced to place some details in the appendix. If accepted, in the camera-ready version, which allows an additional page, we will incorporate key results from the appendix into the main text.
>
> We discuss ablation studies on page 8 in the last paragraph and refer readers to the corresponding figures and tables in the appendix. We would definitely welcome any suggestions you may have on specific results that should ideally be included in the main text.
>
> 4. __...not mentioned at all in the text, such as Figure 2__
>
> We reference Figure 2 on page 8, second line. We will ensure that all figures and tables are explicitly mentioned in the text to improve readability.
>
> 5. __too many references in the appendix; no anonymous code link__
>
> We believe these references provide comprehensive context and support for our work. We plan to release the code once the blind review period is over.

---

### Decision · Program_Chairs · 2024-07-10

**Decision:**

Accept

**Comment:**

This paper develops a novel method for training LLMs to present natural language item endorsements that explain why a recommender system has selected a particular item. If done well (in particular, if properly grounded and not just confabulated), this could help enhance users' agency. The reviewers agree that the technique is well-motivated and well-implemented, and the results demonstrate that the method developed by the authors is effective as compared with some more simplistic alternatives. The reviewers raised some helpful suggestions, to which the authors responded effectively.